# BRG1 attenuates colonic inflammation and tumorigenesis through autophagy-dependent oxidative stress sequestration

Min Liu[1,2,7], Tongyu Sun[3,7], Ni Li[3], Junjie Peng[4], Da Fu [5], Wei Li [6], Li Li [1,2]* & Wei-Qiang Gao[1,2]*

Autophagy is a central component of integrated stress responses that influences many inflammatory diseases, including inflammatory bowel disease (IBD) and colorectal cancer (CRC). While the core machinery is known, the molecular basis of the epigenetic regulation of autophagy and its role in colon inflammation remain largely undefined. Here, we report that BRG1, an ATPase subunit of the SWI/SNF chromatin remodeling complex, is required for the homeostatic maintenance of intestinal epithelial cells (IECs) to prevent the inflammation and tumorigenesis. BRG1 emerges as a key regulator that directly governs the transcription of *Atg16l1*, *Ambra1*, *Atg7* and *Wipi2*, which are important for autophagosome biogenesis. Defective autophagy in BRG1-deficient IECs results in excess reactive oxygen species (ROS), which leads to the defects in barrier integrity. Together, our results establish that BRG1 may represent an autophagy checkpoint that is pathogenetically linked to colitis and is therefore likely a potential therapeutic target for disease intervention.

---

[1] State Key Laboratory of Oncogenes and Related Genes, Renji-Med X Clinical Stem Cell Research Center, Ren Ji Hospital, School of Medicine and School of Biomedical Engineering, Shanghai Jiao Tong University, Shanghai, China. [2] Med-X Research Institute, Shanghai Jiao Tong University, Shanghai, China. [3] CAS Key Laboratory of Tissue Microenvironment and Tumor, CAS Center for Excellence in Molecular Cell Science, Shanghai Institute of Nutrition and Health, Shanghai Institutes for Biological Sciences, University of Chinese Academy of Sciences, Chinese Academy of Sciences, Shanghai, China. [4] Department of Colorectal Surgery, Fudan University Shanghai Cancer Center, Shanghai, China. [5] Central Laboratory for Medical Research, Shanghai Tenth People's Hospital, Tongji University School of Medicine, Shanghai, China. [6] State Key Laboratory of Stem Cell and Reproductive Biology, Institute of Zoology, Chinese Academy of Sciences, Beijing, China. [7] These authors contributed equally: Min Liu, Tongyu Sun. *email: lil@sjtu.edu.cn; gao.weiqiang@sjtu.edu.cn

Tissue inflammation inevitably results in tissue repair and tumorigenesis. Patients with inflammatory bowel disease (IBD), such as Crohn's disease (CD) and ulcerative colitis (UC), exhibit a higher risk of developing colorectal cancer (CRC)[1,2]. IBD results from a complex series of interactions between the mucosal barrier, the environment, and the immune system[3,4]. Barrier integrity is essential for preventing invasion of micro-organisms and development of chronic inflammations[3,5]. Thus, the defects in maintaining the homeostasis of intestinal epithelial cells (IECs) are known to trigger chronic inflammation and repair, which occasionally results in dysplasia[6–8].

The integrated stress response (ISR) serves as a homeostatic mechanism that responds to and resolves various pathogenic insults in the gut[9]. Autophagy represents an intracellular lysosomal degradation system that ensures energetic homeostasis during metabolic stress and plays a protective role by removing damaged components from the cytosol[10–12]. The insufficient autophagy results in the accumulation of undesirable components that fuel inflammation, stimulate ROS, trigger cell death, and induce genomic instability, all of which cause the formation of the hallmarks of cancer[10,13,14]. In IECs, autophagy is implicated in intracellular bacterial killing, antimicrobial peptide secretion by Paneth and goblet cells, and the endoplasmic reticulum (ER) stress response in enterocytes[6,11,15–17]. Likewise, experiments using genetically engineered mouse models (GEMs) indicate that autophagy dysfunctions, such as ablation of *Atg16l1*, *Atg5*, or *Atg7*, stimulate immune responses, ROS production and ER stress[6,11,18]. Importantly, genome-wide association studies (GWAS) have uncovered single nucleotide polymorphisms in gene loci, including the autophagy-related proteins ATG16L1 and IRGM and the nucleotide-binding oligomerization domain-containing protein NOD2, which are closely associated with the risk of IBD[19–21]. Although autophagy is mainly seen as a cytoplasmic event, recent studies indicate that transcriptional and epigenetic regulation occurring in the nucleus intimately modulates autophagy[22,23].

The SWI/SNF complex is a chromatin-remodeling complex that utilizes ATP hydrolysis to affect chromatin utilization[24,25]. The central ATPase in SWI/SNF complexes is either BRG1 (brahma-related gene 1; SMARCA4) or BRM (brahma; SMARCA2)[24]. BRG1 is characterized by a bromodomain and helicase/ATPase activity and influences a plethora of biological processes in both normal and neoplastic tissues[24,26,27]. The loss or mutation of BRG1 was initially identified in several cancers, including but not limited to lung cancer, ovarian cancer and

Burkitt's lymphoma[28–30]. However, other studies also report the positive effects of BRG1 on tumor initiation and progression[31–34], suggesting that the roles of BRG1 in tumors depend on the cellular and genetic milieu. In the gut, previous studies indicate that BRG1 is essential for stem cell function in the small intestine, but not colons[35]. Study using mouse models of intestinal cancer (*Apc*[min/+]) have shown that *Brg1* knockout in the small intestine epithelium attenuates Wnt-driven tumor initiations[34]. In addition, early BRG1 loss impairs duodenum crypt-villous formation partially by regulating the Notch signaling[36]. Despite these advances regarding small intestine development and Wnt and Notch signaling regulation, functions of BRG1 in the colons remain largely undefined.

Recent work using targeted sequencing of UC samples with a high risk of developing colorectal carcinoma indicates that BRG1 is frequently mutated[37], suggesting that BRG1 plays a potential role in inflammatory settings, such as colitis and colitis-CRC transformation. Thus, the present study focuses on determination of the adult function of BRG1 in resolving inflammation in a mouse model of colitis and colorectal tumorigenesis. Using loss- and gain-of-function approaches, we show that BRG1 ensures colonic homeostasis and coupled autophagy-dependent ROS reactions. Thus, our results highlight that BRG1 serves as a homeostatic checkpoint that inhibits inflammation-associated CRC.

## Results

**Epithelial BRG1 expression is reduced in IBD patients.** Consistent with the previous reports, analyses of public datasets suggested that BRG1 mRNA was reduced in IBD specimens as compared with those in healthy controls (using datasets from NCBI's Gene Expression Omnibus: GSE9452 and GSE3365; Fig. 1a). To validate these data in IBD, we performed quantitative RT-PCR (RT-qPCR) assays to determine BRG1 expression in colonic biopsy specimens from CD and UC patients as well as from normal controls. Compared with the levels in biopsies from healthy specimens, we showed that BRG1 mRNA levels were markedly decreased in the IBD biopsies (Fig. 1b). To expand upon these observations, we also performed immunohistochemistry analyses using a pre-valuated BRG1 antibody to characterize the BRG1 expression on the cellular level. The quantification of the immunohistochemical results revealed that the protein levels of BRG1 in the colonic epithelial cells were significantly lower in the IBD specimens relative to that in the healthy subjects

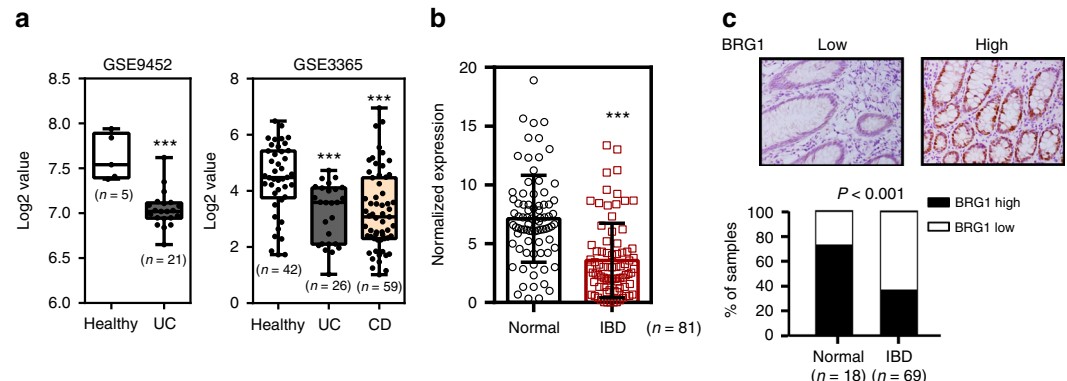

**Fig. 1** BRG1 expression is decreased in IBD patients. **a** Box plot of BRG1 mRNA in healthy controls and IBD specimens (using dataset GSE9452 and GSE3365). In boxplots (middle line depicts the median and the whiskers the min-to-max range). **b** RT-qPCR analysis of BRG1 mRNA in IBD specimens and healthy subjects (*n* = 81 per group). **c** BRG1 staining images are shown in the upper panel, and epithelial BRG1 expressions in normal and IBD biopsies is quantified in the bottom panel ($\chi^2$ test). Staining indexes use a 10-point quantification scale, and a score > 4 is considered higher level. Scale bar: c 50 µm. The data represent the mean ± S.E.M., and statistical significance was determined by a two-tailed Student's *t*-test unless otherwise indicated. ***$p < 0.001$

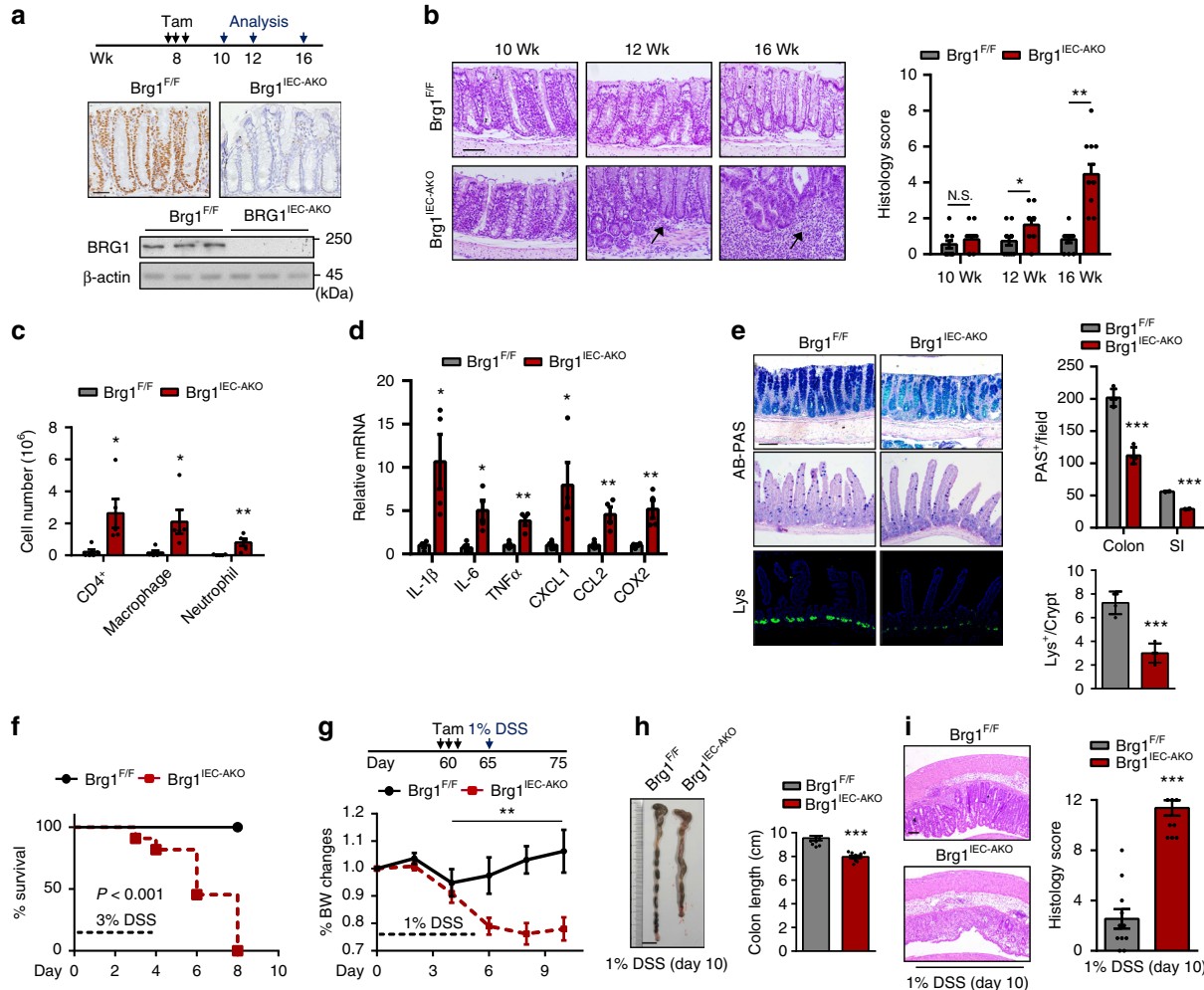

**Fig. 2** *Brg1* loss in adult intestines leads to the development of colitis. Two-month-old mice (*Brg1flox/flox* or *VillinCre-ERT2; Brg1flox/flox*) were treated with tamoxifen, and thereby generated *Brg1F/F* or *Brg1IEC-AKO* mice. **a** Immunohistochemical (upper panel) and immunoblot (IB) analyses (lower panel) of BRG1 expression are shown. **b** Representative histological images of middle-distal sections of *Brg1F/F* and *Brg1IEC-AKO* mice are shown at the indicated time points, and semi-quantitative scoring of the histopathology is shown (*n* = 12 per genotype). Arrow: immune cell infiltration. **c** Colonic lamina propria cells isolated from 16-week-old mice (2 months after the *Brg1* ablation) are analyzed by flow cytometry (*n* = 4 per genotype). **d** RT-qPCR analysis of colon homogenates from 16-week-old mice to assess cytokine and chemokine productions (*n* = 6 per genotype). **e** Alcian blue-Periodic acid Schiff (AB-PAS; goblet cells) staining and lysozyme (Lys; Paneth cells) staining of the intestines derived from 3-month-old mice (1 month after the *Brg1* ablation) and quantitation results are shown in the right (*n* = 5). **f** Survival rate of *Brg1IEC-AKO* mice compared with that of *Brg1F/F* mice after the 3% DSS treatment (*n* = 10). **g-i** Mice were fed 1% DSS in their drinking water and loss of body weights (**g**) and colon length (**h**) are recorded (*n* = 10 per genotype). **i** H&E-stained sections of middle-distal colon tissue collected on day 10 from 1% DSS-treated mice, and the quantitation of histology score are shown in the right. The data represent the mean ± S.E.M., and statistical significance was determined by a two-tailed Student's *t*-test. *$p < 0.05$, **$p < 0.01$, and ***$p < 0.01$. Scale bars: 50 μm (**a**), 100 μm (**b**, **e**, **i**), 1 cm (**h**). N.S., not significant, SI: small intestine

(Fig. 1c). Together, these results suggest a causal link between BRG1 reduction and IBD pathogenesis.

**Adult *Brg1IEC-AKO* mice spontaneously develop colitis.** The above results prompted us to utilize genetically engineered mouse models (GEMs) to define the potential importance of BRG1 in colonic inflammation. To assess the expression pattern of BRG1 in the intestine and colon, we performed co-immunofluorescence staining of BRG1 with Lgr5, ChgA, Muc2 and Lys, respectively, in the small intestine and colon of wild-type mice. BRG1+ cells were located along the intestinal epithelium and BRG1 was co-expressed with Lgr5, ChgA, Muc2 and Lys (Supplementary Fig. 1a), indicating that BRG1 is uniformly expressed both in the basal stem cells and differentiated cells. To circumvent the earlier developmental defects caused by *Brg1* loss, we adopted *VillinCre-ERT2/+* mice[38] to ablate *Brg1* at 2 months of age by

tamoxifen administration (hereafter referred as *Brg1IEC-AKO* mice; deletion of *Brg1* in adult IECs; Fig. 2a). Throughout the studies, *Brg1flox/flox* littermates treated with tamoxifen in the same cages were chosen as the control mice (referred as *Brg1F/F*). We found that the 4-month-old *Brg1IEC-AKO* mice displayed progressive diarrhea, and several mice exhibited rectal bleeding (2 months after the *Brg1* deletion). As compared with *Brg1F/F* mice, 4-month-old *Brg1IEC-AKO* mice exhibited shorter colon lengths and obviously swollen spleen (Supplementary Fig. 1b, c). Further histopathological examinations verified that *Brg1IEC-AKO* mice developed spontaneous colitis, as evidenced by the presence of inflammatory infiltrates, crypt erosion, and the loss of tissue architecture (Fig. 2b). Weekly monitoring and histology quantification throughout the studies revealed no obvious abnormalities within 2 weeks of *Brg1* deletion (Fig. 2b). However, the infiltrated immune cells and mild epithelium erosions were readily detected

after one month of *Brg1* ablation (Fig. 2b). Over time, the 4-month-old *Brg1*[IEC-AKO] mice (2 months after *Brg1* deletion) exhibited severe transmural inflammation affecting the distal colon accompanied by crypt abscesses with nearly 100% penetrance. In contrast, none of the *Brg1*[F/F] littermates housed in the same cages displayed any signs of colonic inflammation. To confirm this observation, we quantified the immune cell infiltration by flow cytometry, and detected a substantial increase in the number of CD4[+] T cells, macrophages and neutrophils in the colonic lysates from the 4-month-old *Brg1*[IEC-AKO] mice (Fig. 2c, Supplementary Fig. 1d). Similarly, the *Brg1*-deficient colons produced significantly higher levels of proinflammatory cytokines and chemokines than the colons of the control littermates (Fig. 2d). The excess inflammatory response occurring in the *Brg1*-deficient mice was associated with a loss of mucosa-producing goblet cells (colon and small intestine) and anti-microbial peptide (AMP)-producing Paneth cells (small intestine) (Fig. 2e, Supplementary Fig. 2e), which contribute to intestinal antibacterial defense by releasing antimicrobial factors. Further course studies revealed that 2 weeks of *Brg1* ablation did not lead to appreciable changes in terms of the colonic stem cells or terminally differentiated cells (Supplementary Fig. 1f–i). However, the 12-week-old *Brg1*[IEC-AKO] mice began to exhibit a decrease in the number of Lgr5[+] cells, goblet cells and Paneth cells (Supplementary Fig. 1g–i). Notably, the average life span of Paneth cells has been estimated to be ~60 days[39], arguing that the reduced number of Paneth cells after 1 month of *Brg1* depletion is likely due to the self-renewal defects. In addition, the numbers of other terminally differentiated cells, such as enteroendocrine cells and tuft cells in 12-week-old *Brg1*[IEC-AKO] mice were not significantly different in comparison to those in the control mice (Supplementary Fig. 1j, k)

To further define the role of BRG1 in colitis, we assessed the consequence of *Brg1* loss in acute colitis by challenging the mice with dextran sodium sulfate (DSS), which is a chemical that induces experimental colitis with the clinical features of IBD. To this end, 2-month-old *Brg1*[F/F] and *Brg1*[IEC-AKO] mice were subjected to 3% DSS treatment, and susceptibility was subsequently monitored. We noticed that *Brg1*[IEC-AKO] mice exhibited hyper-susceptibility to DSS-induced colitis, and none of the *Brg1*[IEC-AKO] mice survived longer than 9 days of observations (Fig. 2f). Therefore, the mice were switched to a lower dose of DSS (1%) challenge. While the *Brg1*[F/F] mice did not display any observable phenotype changes or significant weight loss, the *Brg1*[IEC-AKO] mice exhibited severe colitis, as reflected by weight loss, colon shortening, a loss of goblet cells and Paneth cells, epithelial erosions and inflammatory response (Fig. 2g–i; Supplementary Fig. 2a, b). Meanwhile, *Brg1* ablation did not lead to appreciable changes in terms of enteroendocrine cells and tuft cells (Supplementary Fig. 2c, d). Together, these results highlight that adult BRG1 plays a critical role in resolving inflammatory insults in the colonic epithelium.

**Brg1 loss drives inflammation-associated CRC.** The observation that *Brg1*[IEC-AKO] mice suffered from the sustained inflammation, prompted us to investigate the role of BRG1 in colitis-associated tumorigenesis. Therefore, we first treated the mice with a single dose of the DNA-methylating agent azoxymethane (AOM) to determine whether it enables the *Brg1*[IEC-AKO] mice to develop tumors. Five months after AOM treatment, the *Brg1*[F/F] mice did not grow any lesions as expected; however, all 7-month-old *Brg1*[IEC-AKO] mice developed at least two polyps within the colons (Fig. 3a). The histological examinations verified the presence of epithelium dysplasia in all ten *Brg1*[IEC-AKO] mice examined. Increased tumor incidences in *Brg1*[IEC-AKO] mice were associated

with the persistent inflammation, as reflected by increased IL-6, IL-1β, TNFα and CCL2 productions in the 5 months of AOM-treated *Brg1*[IEC-AKO] mice (Fig. 3b). Together, these results highlight the notion that the chronic inflammation occurring in the *Brg1*[IEC-AKO] mice promotes epithelial dysplasia.

To substantiate this observation, we next induced CRC by injecting AOM, followed by three cycles of 1% DSS treatments. The changes in body weight were recorded throughout the duration of the DSS treatment, and the tumor burden was determined after 12 weeks of AOM treatment. As shown in Fig. 3c, the *Brg1*-deficient mice suffered from chronic inflammation and lost significantly more body weight than the *Brg1*[F/F] mice. Due to 1% DSS treatment, 4 out of 10 *Brg1*[F/F] mice did not develop any polyp, however, all *Brg1*-deficient mice at the same condition exhibited 100% penetrance of tumor development. Compared with the *Brg1*[F/F] mice, the polypoid lesions developed in *Brg1*[IEC-AKO] mice showed a threefold increase in the number, which were on average fourfold larger in size (Fig. 3d). Histological analysis revealed that the lesions developed in *Brg1*[F/F] mice were mainly graded as hyperplasia or low-grade dysplasia. Nevertheless, around 50% of polyps in *Brg1*-deficient colons were classified as high-grade dysplasia (Fig. 3e). Then, we analyzed cell proliferation and apoptosis in these tumors. Proliferation determined by Ki67[+] cells, was slightly increased in the tumors in the *Brg1*[IEC-AKO] mice, whereas apoptotic cells (cleaved caspase-3 positive) did not differ significantly from those in the control mice (Fig. 3f). Thus, adult *Brg1* ablation is strongly detrimental to the development and progression of inflammation-associated CRC, which is different from a previously reported Wnt-activated murine small intestinal model[34].

**BRG1 overexpression protects mice from colitis and CRC.** Next we aimed to determine whether the elevated BRG1 expression could intimately protect the mice from colitis. Thus, we generated conditional BRG1-overexpressing mice (*R26*[BRG1/+]), which harbor a mini gene consisting of a CAGGS (a hybrid chicken β-actin and cytomegalovirus) promoter, a loxP-STOP-loxP (LSL) cassette and flag-tagged BRG1 cDNA knocked in the Rosa26 Locus (Fig. 4a). As characterized by immunohistochemistry and western blotting analyses, a three- to fourfold increase of BRG1 expression in IECs was achieved by the *Villin*[Cre/+]-dependent recombination (*Villin*[Cre/+]; *Rosa26*[Brg1/+], hereafter referred as *Brg1*[IEC-OE/+]; Fig. 4b). During the 6-month follow-up, the *Brg1*[IEC-OE/+] mice did not exhibit detectable abnormalities. In the intestines, the stem/progenitor cells and terminally differentiated cells (goblet cells, enteroendocrine cells and Paneth cells) displayed similar abundances between the *R26*[Brg1/+] (control) and *Brg1*[IEC-OE/+] mice (Supplementary Fig. 3a, b).

To induce acute inflammation, we challenged the mice with 3% DSS. Based on the weight loss, colon length and rectal bleeding, the *Brg1*[IEC-OE/+] mice were more resistant to DSS-induced colitis as compared with *R26*[Brg1/+] mice (Fig. 4c, d). As expected, histological analyses verified that there were fewer areas of ulceration and decreased lymphocyte infiltration in the BRG1-overexpressing colons (Fig. 4e). Similarly, the induction of proinflammatory cytokines and chemokines was markedly reduced in the 3% DSS-treated *Brg1*[IEC-OE/+] mice relative to that in *R26*[Brg1/+] mice (Fig. 4f). The weak inflammatory response occurring in the *Brg1*[IEC-OE/+] mice was also associated with the increased numbers of goblet cells (colon and small intestine) and Paneth cells (small intestine) (Fig. 4g, Supplementary Fig. 3c). Importantly, we found that BRG1 overexpression rendered the mice to become more resistant to AOM/DSS-induced CRC. The pathological quantification verified that the number and size of tumors were markedly reduced in BRG1-overexpressing colons

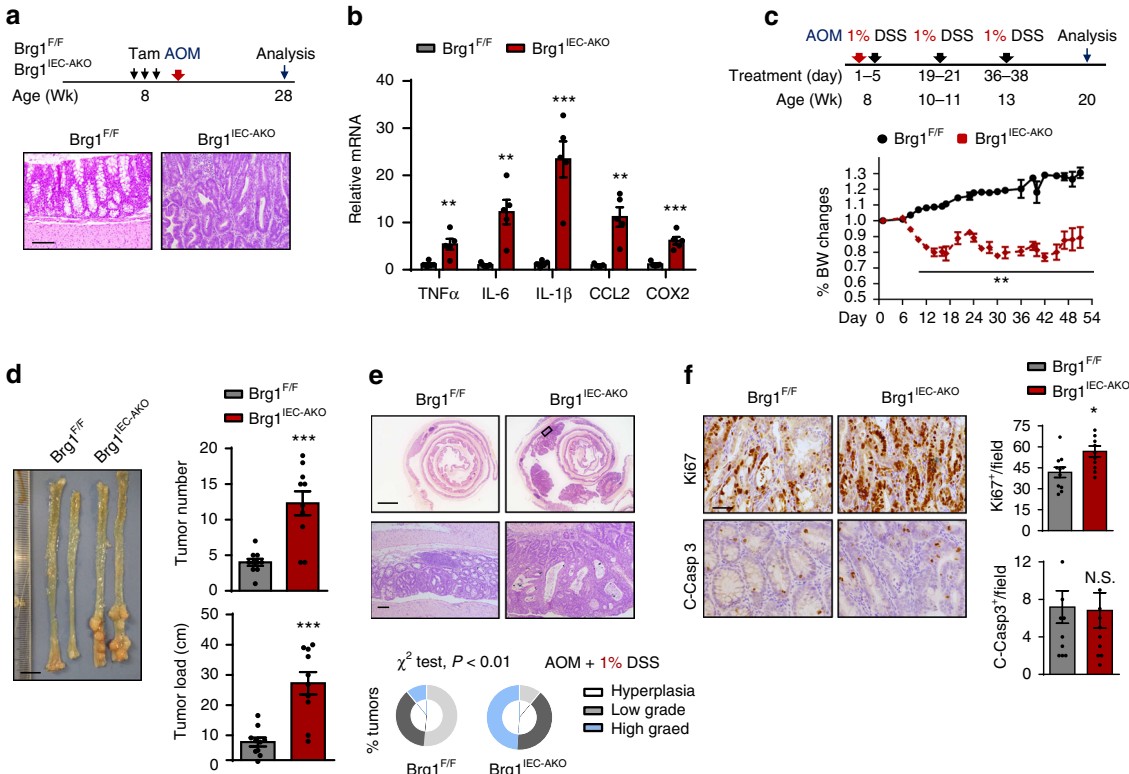

**Fig. 3** *Brg1* loss promotes the inflammation-associated CRC. **a** Scheme of AOM alone treatment protocol. Representative images of H&E-stained colonic sections from *Brg1*[F/F] and *Brg1*[IEC-AKO] mice at 7 months of age (*n* = 10 per genotype). **b** RT-qPCR analysis of the relative mRNA levels of the indicated genes in whole colonic homogenates from 7-month-old mice (*n* = 5 per genotype). **c** *Brg1*[F/F] and *Brg1*[IEC-AKO] mice were injected with AOM on day 0 (2 months of age) and treated with 1% DSS during three 4-day cycles as indicated. Body weight changes are recorded as indicated (*n* = 10 per genotype). **d** Twelve weeks after the AOM injection, the mice were sacrificed to examine the tumor burden (*n* = 10 per genotype). **e** Representative images of colons and the overall grading of the tumors in each genotype (χ² test) are shown (*n* = 10 per genotype). **f** Representative images of Ki67 and cleaved caspase-3 staining in the tumors from 20-week-old AOM/1% DSS-treated *Brg1*[F/F] and *Brg1*[IEC-AKO] are shown (*n* = 4 per genotype). The data represent the mean ± S.E.M., and statistical significance was determined by a two-tailed Student's *t*-test unless otherwise indicated. *$p < 0.05$, **$p < 0.01$, and ***$p < 0.01$. Scale bars: 100 μm (**a**), 50 μm (bottom **e**, **f**), 1 cm (**d**, upper **e**). *N.S.*, not significant

(Fig. 4h). Notably, following 3% DSS treatment, around 30% polypoid lesions developed in *R26*[Brg1/+] mice were characterized as high-grade dysplasia. In sharp contrast, *Brg1* overexpression largely curtailed the tumor development to hyperplasia and low-grade dysplasia (Fig. 4i). Together, these results establish that BRG1 overexpression protects the mice from DSS-induced epithelial damage and subsequent tumorigenesis.

**BRG1 mediates ROS homeostasis to modulate barrier integrity.** Considering that intestinal barrier dysfunctions frequently contribute to gut inflammation[7,8,40], we investigated whether *Brg1* loss compromised the barrier integrity by examining the distribution or expression of tight junction protein 1 (ZO-1), Claudin-1, and E-cadherin before the onset of the severe colitis (3-month-old mice examined). We assessed the distribution of the tight junction protein 1 (ZO-1) in *Brg1*[F/F] and *Brg1*[IEC-AKO] mice as a marker of tight junction structure. In contrast to the control specimens, the *Brg1*-deficient colons exhibited partially disrupted or discontinuous ZO-1 staining (Fig. 5a), indicating that an intact and functional mucosal barrier was compromised in *Brg1*[IEC-AKO] mice. In addition, expressions of other junction proteins, such as Claudin-1, E-cadherin, α-tubulins, and F-actin were also reduced in *Brg1*[IEC-AKO] mice (Fig. 5a, Supplementary Fig. 4a). These findings suggested that *Brg1* ablation resulted in colonic leakage, as evidenced by the enhanced fluorescence in the serum of 3-month-old *Brg1*[IEC-AKO] mice fed with FITC-labeled

dextran (4000 MW) (Fig. 5b). Thus, *Brg1* silencing in adult colons leads to barrier disruption and colonic leakage.

Because epithelial cell death is among the major factors contributing to barrier integrity[4,7,40,41], we therefore assessed the possible defects of cell survival in the *Brg1*[IEC-AKO] mice. Two weeks after tamoxifen treatment, *Brg1* ablation (10-week-old mice) did not lead to appreciable changes in terms of cell death and proliferation (Supplementary Fig. 4b). However, as compared with the *Brg1*[F/F] mice, the numbers of TUNEL- and cleaved caspase-3-positive cells were significantly higher in the colonic sections of the 3-month-old *Brg1*[IEC-AKO] mice (Fig. 5c). Meanwhile, the number of Ki67-positive cells was slightly increased in the 3-month-old *Brg1*[IEC-AKO] mice, presumably due to ongoing regeneration compensated by epithelial proliferation (Fig. 5c). To prove that BRG1 directly influenced the cell survival, we isolated the IECs from the colons of *Brg1*[flox/flox] and *Villin*[Cre-ERT2]; *Brg1*[flox/flox] mice, followed by 4-Hydroxytamoxifen (4-OHT) treatment to delete *Brg1*. The cells were referred as *IEC*[Brg1-F/F] and *IEC*[Brg1-F/F; CreERT2], respectively. After 4 days of *Brg1* deletion, we observed that *Brg1* loss led to a profound increase at the expression levels of cleaved caspase-3 (C-C3) and cleaved PARP (Fig. 5d). Conversely, the apoptotic signal in the IECs isolated from the 5 days of DSS-treated *Brg1*[IEC-OE/+] mice (*IEC*[Brg1-OE/+]) was significantly lower relative to that in the control cells (*IEC*[Brg1/+]) (Fig. 5d). We also established intestinal organoid cultures to further validate these results. As revealed by the 7-aminoactinomycin D (7-AAD) staining and cleaved caspase-3

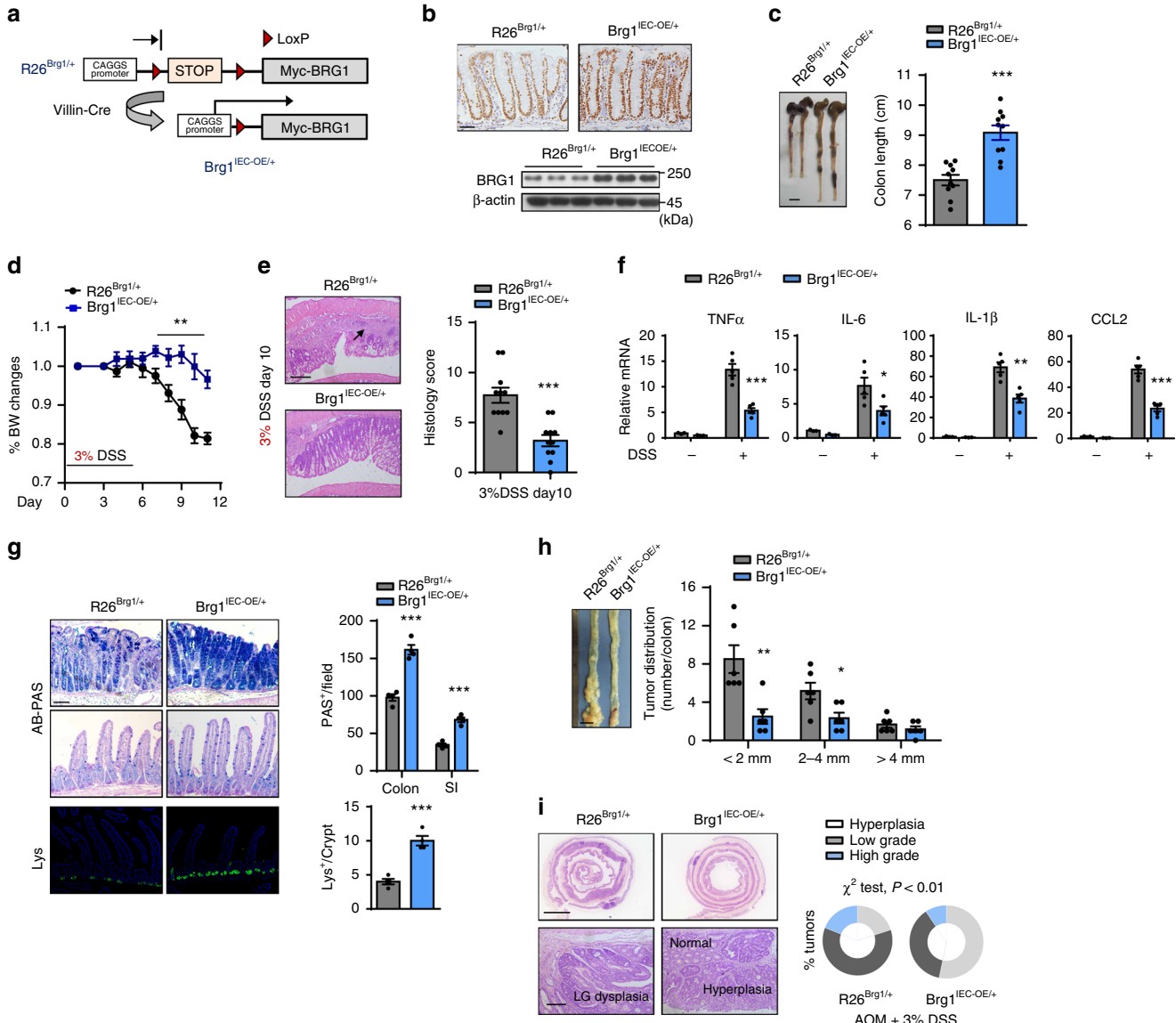

**Fig. 4** BRG1 overexpression in IECs protects the mice from colitis and tumorigenesis. **a** Scheme of $R26^{Brg1/+}$ mice and conditional overexpression of BRG1 in IECs ($Brg1^{IEC-OE/+}$) mice. **b** Immunohistochemistry and western blotting analysis of BRG1 expression in colonic tissues of $R26^{Brg1/+}$ and $Brg1^{IEC-OE/+}$ mice. **c**, **d** DSS was administered for 5 days, and the colon lengths (**c**) and body weights (**d**) are recorded ($n = 10$ per genotype). **e** Representative H&E-stained middle-distal colon sections and histology score (right) from $R26^{Brg1/+}$ and $Brg1^{IEC-OE/+}$ mice treated with 3% DSS ($n = 10$ per genotype). The arrow indicates immune cell infiltration. **f** RT-qPCR analysis of the relative mRNA levels of the indicated genes in whole colonic homogenates from untreated or DSS-treated mice (day 5, $n = 4$ per genotype). **g** Alcian blue-Periodic acid Schiff (AB-PAS; goblet cells) staining and lysozyme (Lys; Paneth cells) staining of the intestines derived from 3% DSS-treated mice and quantitation results are shown in the right ($n = 5$). The 2-month-old mice were treated with AOM, followed by three cycles of 3% DSS treatment. **h** Macroscopic images of the mice after 3 months of AOM treatment, and the numbers the tumors based on the sizes are quantified in the right panel ($n = 10$ per genotype). **i** Representative images of H&E-stained colon sections as indicated; the percent grading of tumors is shown in the right ($n = 10$, $\chi^2$ test). The data represent the mean ± S.E.M., and statistical significance was determined by a two-tailed Student's $t$-test unless otherwise indicated. $*p < 0.05$, $**p < 0.01$, and $***p < 0.01$. Scale bars: 50 μm (**b**, **g**), 200 μm (**e**, bottom **i**), 1 cm (**c**, **h**, upper **i**)

immunostaining, the organoids lacking BRG1 displayed a substantial increase in apoptosis relative to the controls (Fig. 5e, f). Besides, the number of Ki67-positive cells was increased in organoids lacking BRG1 (Supplementary Fig. 4c). These results indicate that BRG1 is intrinsically required for the survival of IECs, whose defects might cause the barrier disruption and the subsequent inflammation.

To understand the mechanistic role of BRG1 in colonic homeostasis, we conducted an expression profile analysis using the IECs isolated from 7-week-old $Brg1^{F/F}$ and $Brg1^{IEC-AKO}$ mice, a time point at which $Brg1$ deletion did not led to colitis yet. Thus,

expression alterations might reflect the primary effects of $Brg1$ loss in colons. Gene Ontology (GO) term analysis of the expression profile indicated no prominent changes of the genes related to apoptotic regulation. However, there was a significant enrichment of genes linked to oxidoreductase activity (Fig. 5g). To validate this finding, we stained 8-oxo-2′-deoxyguanosine (8-OHdG) to examine oxidative stress in colonic tissue and organoids, and detected that 8-OHdG level was indeed increased in the colons/organoids of 7-week-old $Brg1^{IEC-AKO}$ mice (Fig. 5h, Supplementary Fig. 4c). Oxidative stress is associated with CRC pathogenesis, and ROS plays an important role in apoptosis

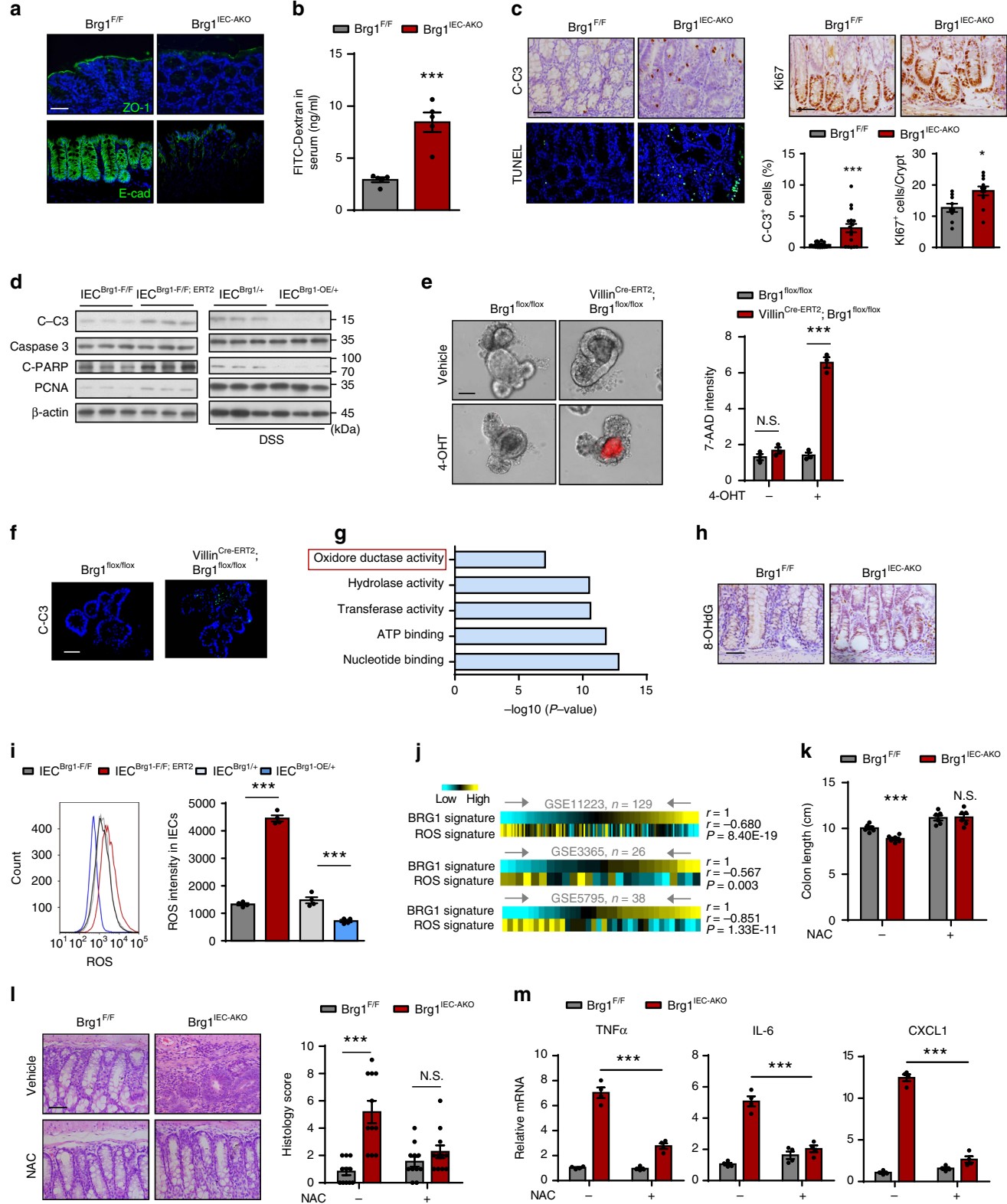

induction under both physiologic and pathologic conditions[6,42]. Therefore, we studied the role of ROS in BRG1-mediated inflammation by staining using 2′,7′-dichlorodihydrofluorescein diacetate (H2DCFDA), which is a fluorescent probe that reacts with ROS. To demonstrate a cell autonomous role of *Brg1* in the ROS homeostasis, the IECs were isolated from *Brg1flox/flox*, *VillinCre-ERT2; Brg1flox/flox* or *R26Brg/+, Brg1IEC-OE/+* mice, and *Brg1* was similarly deleted by 4-OHT treatment. Flow cytometry

analysis indicated that *Brg1*-deleted IECs (*IECBrg1-F/F; CreERT2*) produced significantly more amount of ROS relative to that in the *IECBrg1-F/F* cells, whereas the BRG1-overexpressing IECs (*IECBrg1-OE/+*) exhibited restrained ROS production (Fig. 5i). Similar to what we observed in the mice, the negative associations between BRG1 and the ROS signature were identified in the IBD clinical specimens (using datasets GSE11223, GSE3365, and GSE5795; Fig. 5j). Next, to clarify whether ROS elevation indeed

**Fig. 5** BRG1 mediates ROS reaction to control cell apoptosis and colon inflammation. **a–c** Samples are all derived from 12-week-old $Brg1^{F/F}$ and $Brg1^{IEC-AKO}$ mice (1 month after the tamoxifen injection). **a** Representative ZO-1 and E-cadherin staining in colon sections as indicated. **b** Colonic permeability was measured by the concentration of FITC-dextran in the blood serum ($n = 5$ per genotype). **c** TUNEL, cleaved caspase-3 and Ki67 staining as indicated ($n = 4$). **d** IECs are isolated from $Brg1^{flox/flox}$ and $Villin^{Cre-ERT2}$; $Brg1^{flox/flox}$ or 5-day DSS-treated $R26^{Brg1/+}$ and $Brg1^{IEC-OE/+}$ mice, and Brg1 deletion is achieved by 4-OHT treatment in cells. The corresponding are named as $IEC^{Brg1-F/F}$, $IEC^{Brg1F/F-CreERT2}$ or $IEC^{Brg1/+}$, $IEC^{Brg1-OE/+}$. **e** Organoids derived from the $Brg1^{flox/flox}$ and $Villin^{Cre-ERT2}$; $Brg1^{flox/flox}$ mice. After 5 days of culturing, the organoids were treated with or without 4-OHT, and 7-AAD-stained organoids (red) were imaged after 48 h (left). Quantitation of the fluorescence density per organoid (right). **f** C-Casp3 staining of the organoid sections as indicated. **g** Go term analysis of gene expression changes in the IECs from 7-week-old $Brg1^{F/F}$ and $Brg1^{IEC-AKO}$ mice (1 week after the tamoxifen injection). **h** 8-OHdG staining from 7-week-old $Brg1^{F/F}$ and $Brg1^{IEC-AKO}$ mice. **i** Histograms and MFI quantification of ROS in IECs isolated from the mice as indicated, and Brg1 deletion is achieved by 4-OHT treatment ($n = 4$ per genotype). **j** Correlation (by Pearson's) between BRG1 signaling and ROS signature in the IBD specimens. **k–m** The 12-week-old $Brg1^{F/F}$ and $Brg1^{IEC-AKO}$ mice were treated with NAC after 1 month of Brg1 deletion, and analyzed 1 month later. Colon length (**k**), H&E (**l**), and RT-qPCR analysis (**m**) as indicated ($n = 5$ per genotype). The data represent the mean ± S.E.M., and statistical significance was determined by a two-tailed Student's $t$-test unless otherwise indicated. $*p < 0.05$; $***p < 0.001$. Scale bars: 50 μm (**a**, **c**, **h**, **l**). 20 μm (**f**). N.S., not significant

led to apoptosis of Brg1-deficient IECs, we performed rescue assays by treatment of the Brg1-deficient IECs with an antioxidant, N-acetyl-l-cysteine (NAC). We found that NAC treatment reversed the cleaved caspase-3 signal to the levels similar to the control cells (Supplementary Fig. 4d), arguing that excess ROS indeed contributes to the survival defects of Brg1-deficient IECs.

To provide additional evidence that ROS is a necessary player in BRG1-mediated colitis, we conducted experiments to determine whether blocking ROS could reverse the inflammatory phenotype elicited by the Brg1 loss. To this end, 3-month-old $Brg1^{F/F}$ and $Brg1^{IEC-AKO}$ mice were treated with NAC for 1 month after the Brg1 was deleted at 2 months of age. As judged by colon length and H&E staining (Fig. 5k, l), the blockade of ROS via the administration of NAC in the $Brg1^{IEC-AKO}$ mice completely eased the severity of colitis, and the colonic histology was restored to the normal. Based on the expression levels of proinflammatory cytokines and chemokines, we found that the excess inflammatory response occurring in the $Brg1^{IEC-AKO}$ mice was largely alleviated by the ROS inhibition (Fig. 5m). In addition, the NAC treatment replenished the number of goblet and Paneth cells and relieved epithelial apoptosis to levels similar to those in the control mice (Supplementary Fig. 4e–g). Collectively, these data indicate that ROS plays a key role in BRG1-mediated colitis.

**BRG1 serves as an autophagy checkpoint in IECs**. To elucidate the molecular basis by which BRG1 modulated ROS, BRG1-bound chromatin from 2-month-old wild-type IECs was immunoprecipitated and analyzed by performing deep sequencing. The ChIP-seq analysis revealed that 12119 genes (26349 peaks) possessed BRG1 occupancies within 6 kb of annotated genes. The BRG1 binding intervals were located within the core transcribed portion of genes close to the transcription start site (TSS) (Supplementary Fig. 5a, b). To correlate the chromatin binding with the transcriptional regulation, the ChIP-seq data were aligned with the expression profile. The Venn diagrams indicated that 746 genes showed direct BRG1 occupancies and an expression downregulation upon Brg1 ablation (Fig. 6a). Among these genes, we noticed that the process of Autophagic vacuole fusion, Autophagic vacuole assembly and Autophagy was significantly enriched. As summarized by the heatmap, expression levels of central autophagy-regulatory genes, including Atg16l1, Atg7, Ambra1, Wipi2, were significant reduced in Brg1-deleted IECs (Fig. 6b), suggesting a possible defect of autophagy in the absence of Brg1.

Several lines of evidences indicate that impaired autophagy causes oxidative stress[6,16]. We therefore reasoned that BRG1 modulated ROS homeostasis through regulation of autophagy. To test this possibility, we isolated IECs from the colons of 2-month-

old $Brg1^{flox/flox}$ and $Villin^{Cre-ERT2}$; $Brg1^{flox/flox}$ or $Rosa26^{Brg1/+}$ and $Brg1^{IEC-OE/+}$ mice. After 4 days of culturing, western blotting analysis revealed that the BRG1-depleted IECs (by treatment of 4-OHT) exhibited a profound reduction in membrane-bound LC3-II (LC3B-I to LC3B-II conversion) and an increase in the cellular levels of the autophagic substrate p62 (Fig. 6c). Conversely, IECs bearing BRG1 overexpression displayed an enhanced LC3-I/II conversion and a simultaneously decreased p62 level (Fig. 6c). In addition, the impaired autophagy was also supported by intestinal organoid culture assay. As reflected by L3II conversion and p62 level, Brg1 deletion led to a reduction of autophagy (Supplementary Fig. 5c). Further transmission electron microscopy assays verified that the number of autophagosomes was reduced in the colons following 1 week of Brg1 deletion as compared with that in controls (Fig. 6d). To ensure this observation, we generated Brg1-deficient autophagic reporter mice ($Brg1^{IEC-AKO}$; LC3-GFP) and detected a significant decrease in the number of LC3-GFP puncta in the colons of 7-week-old $Brg1^{IEC-AKO}$ mice (1 week after Brg1 ablation; Fig. 6e). Importantly, the cell intrinsic requirement of Brg1 to govern autophagy was supported by IEC culture assay. As reflected by the number of LC3 puncta, L3II conversion, and p62 level, Brg1 deletion led to a gradual reduction of autophagy in a time course dependent on Brg1 deletion (Fig. 6f, g). As LC3II accumulation is attributed to either increased autophagy induction or impaired autophagosome turnover, the effects of Brg1 loss on autophagic flux were evaluated in the presence of chloroquine (CQ), an inhibitor of lysosomal degradation. We observed that LC3II levels in the presence of CQ were persistently reduced in Brg1-depleted IECs (Fig. 6h–i). Thus, these results indicate that BRG1 is required for autophagic induction in the colonic epithelium. Next, we aimed to clarify whether defects of autophagy indeed contributed to uncontrolled ROS reactions in Brg1-deficient IECs. Treatment with rapamycin to enhance autophagy in Brg1-depleted IECs upregulated the expression levels of the key autophagy regulators, and led to the levels of ROS and cell death comparable to those observed in the $IEC^{Brg1-F/F}$ cells (Supplementary Fig. 5d–g), suggesting that Brg1 loss impairs autophagy and thereby enhances ROS productions.

To distinguish between BRG1 binding sites and BRG1 functional sites genome-wide, we performed H3K9ac (a histone marker of active gene transcription) ChIP-seq in $Brg1^{F/F}$ and $Brg1^{IEC-AKO}$ IECs. The ChIP-seq analysis revealed that 12573 genes possessed H3K9ac occupancies within 6 kb of annotated genes (Supplementary Fig. 5h). The Venn diagrams indicated that there were still 685 genes overlapped (Fig. 6j). GO term analysis identified Autophagic vacuole fusion and Autophagic vacuole assembly pathways enriched in the 685 genes (Supplementary Fig. 5i). To define the regulatory mechanism by which BRG1 controlled autophagy, we referenced the express profiles and ChIP-seq results, and examined expression levels of key

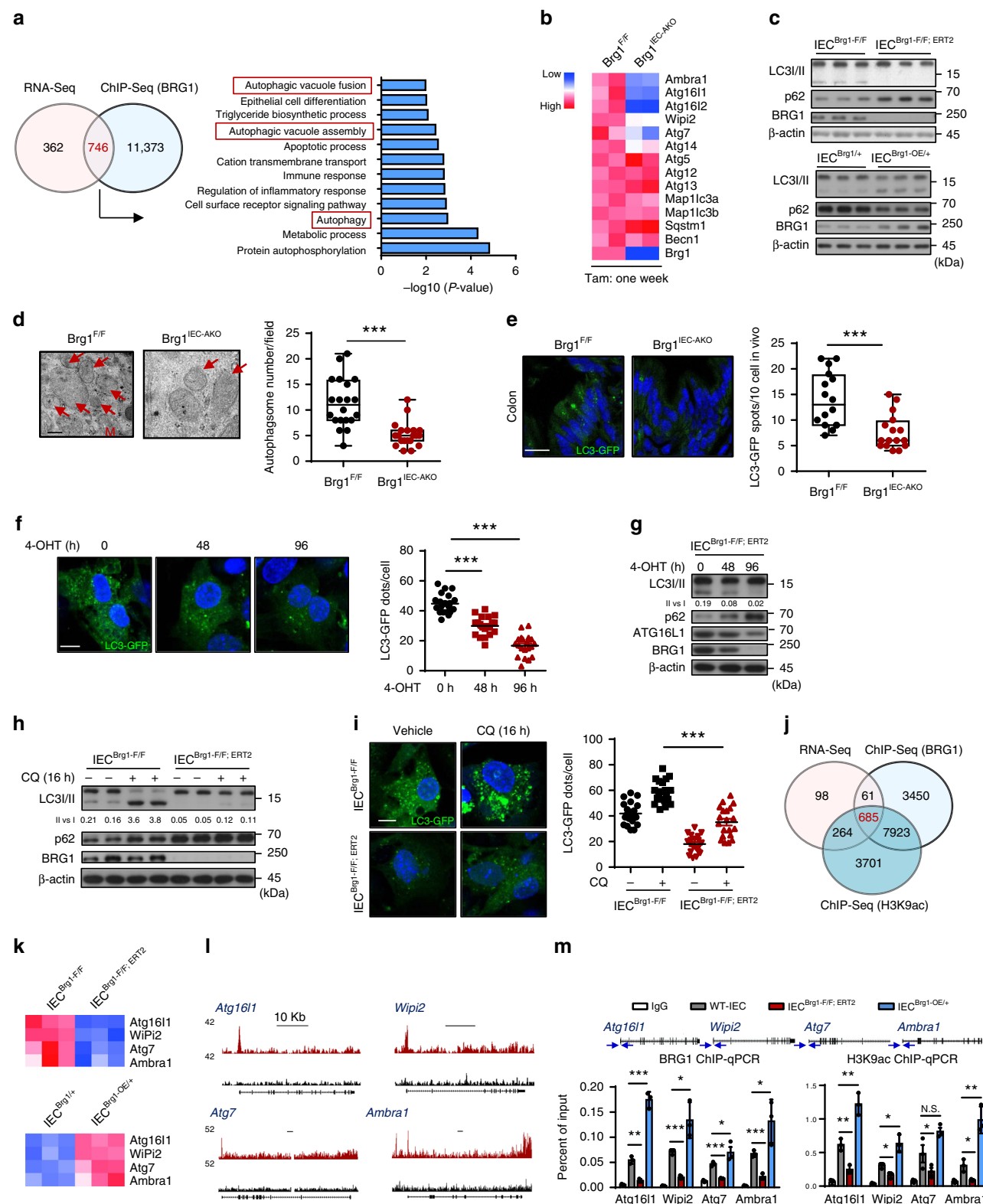

autophagy-regulatory genes including *Atg16l1*, *Wipi2*, *Atg7*, and *Ambra1*. As expected, BRG1 loss and overexpression indeed down- and upregulated their expression levels in the IECs, respectively (Fig. 6k). Direct BRG1 and H3K9ac occupancies within the candidate gene loci were also seen in Genome Browser tracks (Fig. 6l, Supplementary Fig. 5j) and readily validated by the ChIP-qPCR assays (Fig. 6m). Consistent with the decreased BRG1 recruitment to these gene loci, H3K9ac was simultaneously reduced in the *Brg1*-depleted IECs (Fig. 6m). Conversely, BRG1 overexpression enhanced the H3K9ac signals within the promoter regions of the *Atg16l1*, *Wipi2*, *Atg7*, and *Ambra1* gene loci (Fig. 6m). Thus, *Brg1* loss led to a repressive chromatin status that directly repressed the transcription of the genes important for autophagosome initiation. Consistent with the GEM results, we found positive associations between BRG1 and ATG16L1, WIPI2, and AMBRA1 in the IBD biopsies (using datasets from

**Fig. 6** BRG1 regulates autophagy in IECs to control ROS and colonic inflammation. IECs were isolated from $Brg1^{flox/flox}$, $Villin^{Cre-ERT2}$; $Brg1^{flox/flox}$ or $R26^{Brg1/+}$ and $Brg1^{IEC-OE/+}$ mice, and $Brg1$ deletion is achieved by 4-OHT treatment in cells (hereafter named as $IEC^{Brg1-F/F}$, $IEC^{Brg1-F/F-CreERT2}$ or $IEC^{Brg1/+}$, $IEC^{Brg1-OE/+}$). **a** Venn diagram showing the number of genes harboring BRG1 binding and displaying expression changes in $Brg1$-KO IECs (1 week of $Brg1$ ablation). Right panel shows the Go term analysis of the overlapping genes. **b** Heatmap summarizes the RNA-seq results of gene expression related to autophagy regulation. **c** Immunoblotting of LC3 conversion, p62 levels as indicated. **d** Autophagosome detection by transmission electron microscopy and the quantification in the colons from 7-week-old $Brg1^{F/F}$ and $Brg1^{IEC-AKO}$ mice (1 week of $Brg1$ deletion; $n = 5$ per genotype). Arrow indicates the Autophagosome, M: Mitochondrion. **e** LC3-GFP staining in the colons from 7-week-old $Brg1^{F/F}$ and $Brg1^{IEC-AKO}$ mice. **f, g** LC3-GFP staining (**f**) and IB analysis of the indicated protein in $IEC^{Brg1F/F-CreERT2}$ treated with 4-OHT treatment as indicated (**g**). **h, i** IB analysis (**h**) and LC3-GFP staining (**i**) in $IEC^{Brg1-F/F}$ and $IEC^{Brg1F/F-CreERT2}$ with or without chloroquine (CQ 10 μM) treatment. **j** Venn diagram indicating overlapping genes with BRG1, H3K9ac binding and displaying expression changes in $Brg1$-KO IECs. **k** RT-qPCR analysis of the indicated genes in the IECs ($Brg1$ KO or overexpression) as indicated, and the results are summarized by heatmap ($n = 3$). **l** Snapshot of BRG1 ChIP-Seq signals at the $Atg16l1$, $Wipi2$, $Atg7$, and $Ambra1$ gene loci in IECs isolated from wild-type mice. **m** ChIP-qPCR analysis of BRG1 binding and H3K9ac codes for the $Atg16l1$, $Wipi2$, $Atg7$, and $Ambra1$ gene loci in the IECs as indicated ($n = 5$ per genotype). The arrow indicates the locations of the ChIP-qPCR primer pairs. The data represent the mean ± S.E.M., and statistical significance was determined by a two-tailed Student's $t$-test. $*p < 0.05$; $**p < 0.01$; $***p < 0.001$. Scale bars: 200 nm (**d**), 50 μm (**e**), 10 μm (**f**), 10 kb (**l**)

GSE57945; Supplementary Fig. 5k). Together, our results indicate that BRG1 modulates the multifaceted autophagy machinery to govern colon homeostasis.

**BRG1 regulates autophagy to control ROS reactions in IECs.** Given the functional and clinical importance of ATG16L1 in autophagosome formations and colitis[11,19], we assessed whether the downregulation of $Atg16l1$ in $Brg1$-deleted IECs indeed caused the uncontrolled ROS reactions and the defects in cell survival. Thus, we used the oligos to knockdown (KD) $Atg16l1$ in $IEC^{Brg1-F/F}$ and $IEC^{Brg1-F/F;CreERT2}$. As expected, $Atg16l1$ ablation in $IEC^{Brg1-F/F}$ greatly reduced the LC3I/II conversion, and meanwhile enhanced ROS and cleaved caspase-3 signals (Fig. 7a, b). Importantly, $Atg16l1$ KD in $Brg1$-deficient IECs did not further reduce the autophagy response. The cells with or without BRG1 ablation exhibited a similar extent of ROS elevation as well as the cleaved caspase-3 level (Fig. 7a, b). Restoration of ATG16L1 in $Brg1/Atg16l1$-depleted cells largely alleviated the overproductions of ROS and improved the cellular survival as compared with $Brg1$-deleted IECs (Fig. 7a, b). Together, these results highlight the requirement of $Brg1$ for the regulation of $Atg16l1$ or autophagy to modulate ROS homeostasis in IECs.

Besides $Atg16l1$, we showed that BRG1 directly regulated $Wipi2$, $Atg7$, and $Ambra1$, which are also important for autophagosome biogenesis. To further clarify the functional dependence between $Brg1$ and autophagy in colonic inflammation, we generated $Villin^{Cre/+}$; $Atg5^{flox/flox}$ mice that conditionally ablated the autophagy protein ATG5 in the IECs ($Atg5^{IEC-/-}$) (Fig. 7c). Upon treatment with 3% DSS, compared with the littermate controls, the $Atg5^{IEC-/-}$ mice exhibited a greater weight loss, enhanced colon length shortening, and immunopathology (Fig. 7d–f), confirming the indispensable role of autophagy in resolving gut inflammation. More importantly, we showed that the $Atg5$ ablation in the $Brg1^{IEC-OE/+}$ mice compromised the protective role of BRG1 overexpression against the DSS treatments. As reflected by the body weight loss, colon length and histological examinations, the $Brg1^{IEC-OE/+}$; $Atg5^{IEC-/-}$ mice exhibited epithelial erosion and immune cell infiltration comparable to those in the $Atg5^{IEC-/-}$ mice (Fig. 7d–f). In addition, depletion of $Atg5$ in the $Brg1^{IEC-OE/+}$ mice greatly enhanced the inflammatory cytokine, 8-OHdG and ROS production to the levels similar as those observed in the $Atg5^{IEC-/-}$ mice (Fig. 7g, h; Supplementary Fig. 6a). Meanwhile, the $Atg5$ ablation in the $Brg1^{IEC-OE/+}$ mice led to severe barrier disruption and epithelial cell death comparable to those in the $Atg5^{IEC-/-}$ mice (Supplementary Fig. 6b–e). Thus, defective autophagy in $Atg5$-deficient mice resulted in excess ROS, which led to the defects in cellular apoptosis and barrier integrity, and the subsequent inflammation. Altogether, the GEMs indicate that $Brg1$ prevents

colon inflammation dependent on the regulation of the autophagy to restrain ROS over-reactions.

The intestinal microbiota plays an important role in driving inflammatory responses during disease development and progression[43–45]. To rule out the possibility that the gut microbiota contributes to intestinal barrier damage caused by BRG1 deficiency, we generated gut microbiota-depleted mice by treating $Brg1^{IEC-AKO}$ and $Brg1^{F/F}$ mice with drinking water containing an antibiotic cocktail for 3 weeks and then treated them with 1%DSS. The depletion of gut microbiota was confirmed by performing 16S rDNA sequencing. The successful depletion of the gut microbiota was evident as no sufficient bacterial diversity and microbiota composition were observed (Supplementary Fig. 7a–c). As compared with the $Brg1^{F/F}$ mice, there was less bacterial diversity but unchanged gut microbiota composition in untreated $Brg1^{IEC-AKO}$ mice (Supplementary Fig. 7a–c). Interestingly, depletion of the gut microbiota by antibiotics reduced expression of proinflammatory cytokines and chemokines of $Brg1^{IEC-AKO}$ mice comparable to those observed in the $Brg1^{F/F}$ mice (Supplementary Fig. 7d). However, antibiotic-treated $Brg1^{IEC-AKO}$ mice still exhibited more severe barrier disruption, epithelial cell apoptosis, impaired autophagy, and oxidative stress than $Brg1^{F/F}$ mice (Supplementary Fig. 7e–i). These results suggested that depletion of the gut microbiota did not facilitate the recovery of intestinal barrier in $Brg1^{IEC-AKO}$ mice. Thus, the gut microbiota does not contribute to intestinal barrier disruption caused by BRG1 deficiency. Altogether, BRG1 protects the colon from inflammatory insults via a mechanism dependent on autophagy and ROS sequestration (Fig. 7i).

## Discussion

Autophagy, a system for sensing and adapting to environmental changes, is intimately involved in cellular defense, microbial tolerance, and metabolic control[10–12]. Despite recent progress, the underlying mechanism by which this degradative process is balanced remains unclear. Here, we showed that multifaceted autophagy-component genes, including $Atg16l1$, $Ambra1$, $Wipi2$, and $Atg7$, are directly controlled by BRG1 in the colons. ATG16L1 and ATG12-ATG5 complex formation is known to define the site for LC3 PE conjugation and facilitate autophagosome formation[11,46]. $Ambra1$ interacts with Beclin1 through the target lipid kinase Vps34/PI3KC3 to assemble a class III PI3K complex, which positively regulates the formation of autophagosomes[47,48]. In addition, WIPI2 is a PtdIns(3)P effector upstream of ATG16L1 that is required for LC3 conjugation[49,50]. Thus, our results demonstrated that BRG1 is a key regulator controlling autophagosome biogenesis and therefore highlight the homeostatic function of BRG1 in colons.

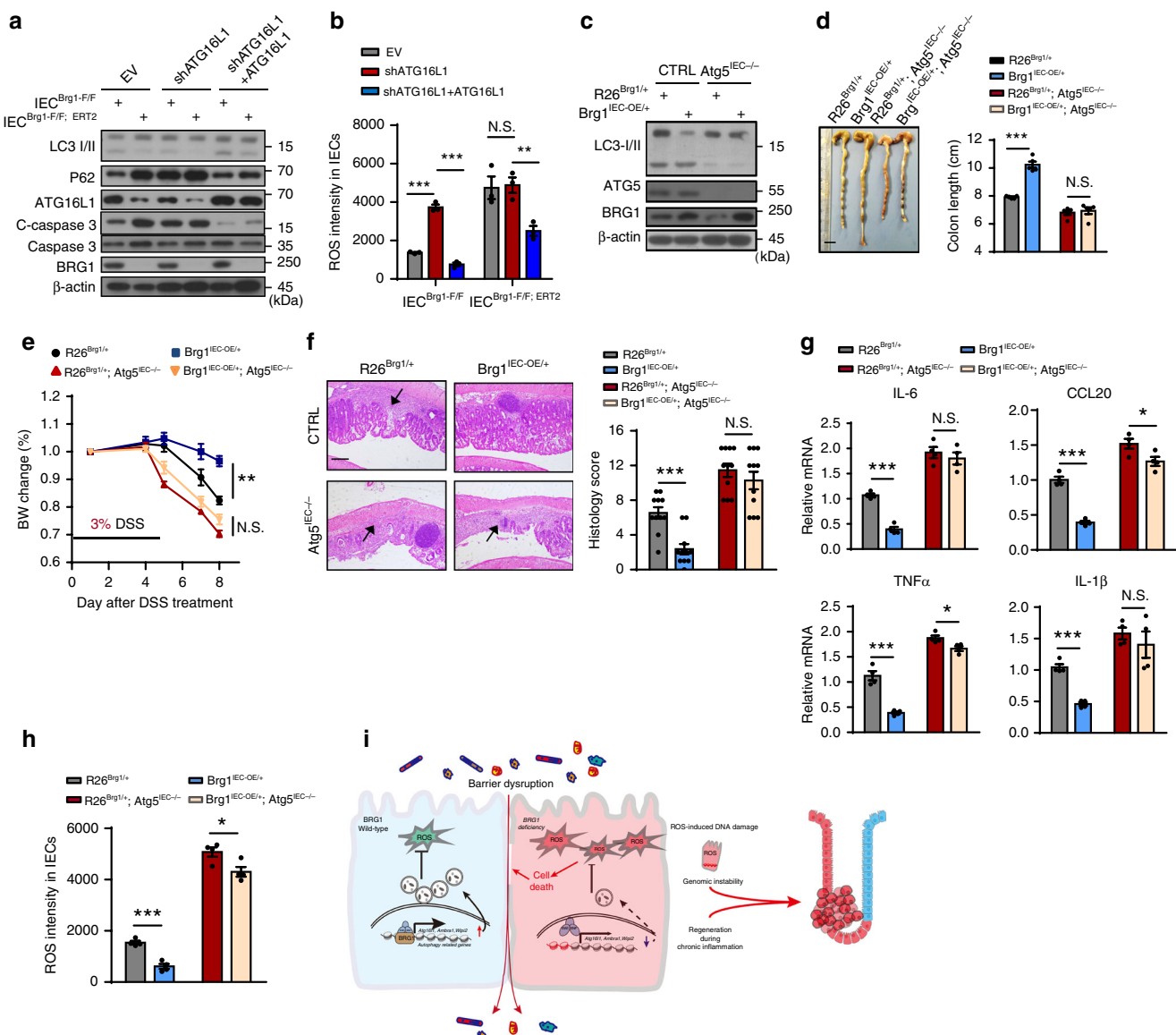

**Fig. 7** BRG1 protects colon inflammation dependent on the regulation of autophagy. **a** IB analysis of the indicated protein in $IEC^{Brg1-F/F}$ (control) or $IEC^{Brg1-F/F-CreERT2}$ (*Brg1* KO) with or without ATG16L1 knockdown or restoration. **b** Quantification of ROS levels in the IECs as indicated ($n = 4$ per genotype). **c** Immunoblotting analysis of BRG1, ATG5, and LC3 in IECs isolated from the mice as indicated. **d–h** DSS was administered in drinking water for 5 days. The colon lengths (**d**) and body weights (**e**) are recorded in the $R26^{Brg1+}$ and $Brg1^{IEC-OE/+}$ mice with or without *Atg5* deletion in IECs ($n = 8$ per genotype). **f** Histology of the colonic sections from the indicated mouse genotypes after 9 days of DSS treatment. **g** RT-qPCR analysis of the relative mRNA expression levels in colonic homogenates from DSS-treated mice (day 9; $n = 5$ per genotype). **h** Quantification of ROS levels in the IECs of mice as indicated ($n = 4$ per genotype). **i** BRG1 directly controls multifaceted autophagy-component genes including *Atg16l1*, *Wipi2*, *Atg7*, and *Ambra1* in IECs. Thus, adult *Brg1* loss in IEC leads to insufficient autophagy, which results in excess reactive oxygen species (ROS), and thereby compromises barrier integrity and promotes inflammation. Compensatory regeneration coupled with ROS-induced DNA damage promotes the malignant progression of CRC. The statistical significance was determined by a two-tailed Student's *t*-test. *$p < 0.05$ and **$p < 0.01$; ***$p < 0.001$. Scale bars: 0.5 cm (**d**), 100 μm (**f**). *N.S.*, not significant

Mammalian cells are consistently exposed to ROS generated from either endogenous metabolism or environmental oxidants[51,52]. We found that *Brg1* loss in adult mice leads to ROS accumulations, and thereby enhances epithelium death and barrier integrity. This finding is further supported by the antioxidant NAC treatment, which attenuates the ROS reactions and relieves the colitis symptoms in the $Brg1^{IEC-AKO}$ mice. Thus, aberrant ROS accumulation is mainly responsible for the chronic inflammation observed in the $Brg1^{IEC-AKO}$ mice. Autophagy has been considered the main mechanism responsible for removing oxidized proteins in cells[11,13]. In the intestines, defects in autophagy, such as *Atg16l1* and *Atg5* ablation,

stimulate ROS reactions[6,11,53]. The current studies utilize GEMs and cell cultures to clarify that BRG1 attenuated colonic inflammation through autophagy-dependent oxidative stress. We showed that *Atg5* depletion in the $Brg1^{IEC-OE/+}$ mice compromises the protective effects elicited by the BRG1 over-expression and enhances the ROS levels to those observed in the $Atg5^{IEC-/-}$ mice. Conversely, restoration of *Atg16l1* or the treatment with rapamycin in the *Brg1*-depleted IECs reduces the levels of ROS and improved epithelial cell survival. Thus, adult BRG1 protects the colon from inflammatory insults via a mechanism dependent on autophagy and ROS sequestration (Fig. 7i).

Unresolved tissue injury during inflammation inevitably results in tissue neoplastic transformation. Our current study demonstrates that adult *Brg1* loss results in compensatory regeneration coupled with ROS-induced DNA damage in an inflammation-context, which promotes the malignant progression of CRC. This finding is further corroborated by the reduction in acute inflammation and tumorigenesis in the BRG1-overexpressing mice. Notably, these results differ from a previous report in which the loss of *Brg1* suppresses adenoma formation in an *Apc*-mutated context[34]. Regarding the opposing tumorigenic functions in these two models, numerous studies have shown that the functions of the mammalian SWI/SNF enzyme appear to be cell type- and subunit-dependent[24,54,55]. Thus, unsurprisingly, BRG1 can act as either an oncogene or tumor suppressor in a context-dependent manner.

BRG1 plays a pivotal role in crypt-villous remodeling, differentiation and cell survival in the duodenum of neonatal mice by regulating the Notch signaling[36]. In the present study, we also examined whether Notch signaling is altered in adult *Brg1*-depleted colons. However, no appreciable alternations are detected in the *Brg1*$^{IEC-AKO}$ mice as determined by the expression levels of the Notch ligands Dll1, Dll3, Dll4, Jag1, and Jag2 and the Notch target genes Hes1, Hey1, and Hey2 (Supplementary Fig. 8). Moreover, BRG1 is known to modulate stem cell self-renewal in the small intestines by regulating the Wnt signaling[35]. We did not observe changes in stem/progenitor cells in the colons before the onset of colitis (Supplementary Fig. 1). In contrast, the numbers of Paneth and goblet cells are reduced in the adult *Brg1*-depleted mice, whereas these defects are reversed by the NAC treatment. Furthermore, the results obtained from the BRG1 overexpression mice further support the idea that BRG1 is intimately involved in colon epithelial cell homeostasis without altering the differentiation process. However, because BRG1 is a key epigenetic regulator involved in multiple transcriptional processes, we cannot fully exclude the possibility that other signaling changes also contribute to the phenotype observed in the present study. In addition, the underlying mechanism leading to reduced BRG1 expression during chronic inflammation remains to be determined in future studies.

Together, our results highlight the importance that BRG1 functions as a homeostatic regulator that integrates the intact mucosal barrier by modulating autophagy-dependent oxidative stress in the colon (Fig. 7i). Inflammation is closely associated with tumorigenesis, and IBD is a predisposing factor for CRC. Thus, our results establish a role for BRG1 in inflammation-associated CRC and may provide insight into understanding of other human disorders associated with BRG1 deficiency.

## Methods

**Human specimen analysis.** Patients with IBD and non-IBD healthy subjects for this study were recruited from Fudan University Shanghai Cancer Center and Shanghai Tenth People's Hospital, China[40]. The study was approved by the ethics committee of the Shanghai Tenth People's Hospital (SHSY-IEC-pap-16-24). The use of pathological specimens as well as the review of all the pertinent patient records was approved by the institutional ethics review board. Immunohistochemical analyses were performed using a specific anti-BRG1 antibody (Abcam, Cat# ab110641 (1:500))[40,56]. In brief, protein expression was scored and quantified by pathologists who were blinded to the outcome of the cases, and the quantification was based on a multiplicative index of the average staining intensity (0–3) and the extent of staining (0–3), which yielded a 10-point staining index that ranged from 0 to 9. Tissue RNA from colonic biopsies of IBD patients and heathy controls was also prepared for the analysis of gene expression.

**Mice.** All mice were maintained in a specific-pathogen-free (SPF) facility and all experimental procedures were approved by the institutional biomedical research ethics committee of the Shanghai Institutes for Biological Sciences or Institute of Zoology, Chinese Academy of Sciences. *Brg1*$^{IEC-AKO}$ mice were generated by crossing *Brg1*-floxed mice[57] with *Villin*$^{Cre-ERT2}$ mice (The Jackson Laboratory)[38], and tamoxifen was intraperitoneally injected for 3 consecutive days at 100 mg kg$^{-1}$

body weight. BRG1-overexpressing mice (*Brg1*$^{IEC-OE/+}$) were generated by crossing *R26*$^{Brg1/+}$ mice[56] with *Villin*$^{Cre/+}$ mice. LC3-GFP mice were purchased from The Jackson Laboratory and *Atg5*-floxed mouse strain (RBRC02975)[58] were purchased from the RIKEN Bio Resource Center with permission from Dr. Noboru Misushima (The University of Tokyo). All the mice were maintained on C57BL/6J background and littermates with the same treatment were used for control experiments. For permeability experiments, 3-month-old mice were fasted for 4 h, and treated with FITC-conjugated dextran (500 mg kg$^{-1}$ body weight). The fluorescence intensity was determined by FITC-dextran standard curve.

**Induction of colitis and CRC.** All the mice at 2 months of age were first treated with tamoxifen (100 mg kg$^{-1}$ body weight) for 3 consecutive days to induce *Brg1* deletion, and then received DSS or AOM/DSS treatments as indicated. To induce colitis, mice were fed with 3 or 1% DSS (molecular weight, 36–50 kDa; MP Biomedicals) for 3 or 5 days, followed by regular drinking water, body weight was daily recorded. For AOM alone treatment, *Brg1*$^{F/F}$ and *Brg1*$^{IEC-AKO}$ mice were intraperitoneally injected with 10 mg kg$^{-1}$ AOM (Sigma), and mice were sacrificed 20 weeks later. In AOM/DSS experiments, mice were first treated with a single dose of AOM, and 1 or 3% DSS was given in the drinking water for 3 days, followed by 2 weeks of regular drinking water. The DSS treatment was repeated for two additional cycles, and mice were sacrificed 12 weeks after AOM injection, except when indicated otherwise.

**IEC isolation and culture.** IECs were isolated from *Brg1*$^{flox/flox}$, *Villin*$^{Cre-ERT2}$; *Brg1*$^{flox/flox}$ or *R26*$^{Brg1/+}$ and *Brg1*$^{IEC-OE/+}$ mice, and *Brg1* deletion is achieved by 4-OHT (5 μM) treatment in cells. The IECs corresponding to each genotype were referred as *IEC*$^{Brg1-F/F}$, *IEC*$^{Brg1-F/F; CreERT2}$ or *IEC*$^{Brg1/+}$, *IEC*$^{Brg1-OE/+}$. For RNA-Seq and ChIP-Seq analyses, IECs from colons were isolated using isolation buffer (30 mM EDTA and 1 mM DTT). For culture purpose, the colons were cut into pieces and washed by DMEM for two times, then incubated with digestion buffer (DMEM with collagenase type I and Dispase II) for 2 h in 37 °C. After the incubation, the cell suspension was passed through 100 μm cell strainers (Corning). After washes, the cells were plated in dishes coated with rat tail tendon collagen type I overnight and were cultured in DMEM with 10% FBS and 1% penicillin/streptomycin. The next day, the cells were treated with rapamycin (20 nM) or NAC (5 mM) for subsequent experiments. The purities of colonic epithelial cell were confirmed by FACS analysis of EpCAM staining. For detecting the ROS levels, cultured IECs were digested and centrifuged, the precipitates were resuspended by PBS and incubated with CM-H2DCFDA (5 mM; C6827, Life Technologies) for 10 min in 37 °C and measured by flow cytometry.

**Organoid culture and analysis.** The intestines were opened longitudinally, and villi were scraped away. After thorough washing in cold PBS, the pieces were incubated in 2 mM EDTA/PBS for 10 min at 4 °C. Then, EDTA solution was replaced with PBS and shacked vigorously for 45 s. Crypt fractions were purified by successive centrifugation steps. Hundred microliters of Advanced DMEM/F12 (Invitrogen) containing growth factors (50 ng ml$^{-1}$ EGF, PeproTech; 500 ng ml$^{-1}$ R-spondin, PeproTech and 100 ng ml$^{-1}$ Noggin; PeproTech) was added and refreshed every 2 or 3 days. 4-OHT was added to medium at a concentration 2 μM after 5 days of organoid culturing. After 48 h, the organoids were stained with 7-AAD for 5 min, and photos were imaged and quantified by Image J software.

**RNA-Seq and ChIP-Seq analysis.** IECs was isolated from 7-week-old *Brg1*$^{F/F}$ and *Brg1*$^{IEC-AKO}$ by EDTA-based isolation. Each sample contained three to five independent repeated animals, and subjected to HiSeq performed by Guangzhou RiboBio Co., Ltd. Paired-end reads were aligned to the mouse reference genome mm10 with HISAT2. HTSeq v0.6.0 was used to count the reads numbers mapped to each gene. The whole samples expression levels were presented as RPKM (expected number of Reads PerKilobase of transcript sequence per Million base pairs sequenced), which is the recommended and most common method to estimate the level of gene expression. The statistically significant DE genes were obtained by an *p*-value threshold of <0.05 and fold change ≥1.5. All differentially expressed mRNAs were selected for GO analysis which was performed on DAVID (https://david.ncifcrf.gov/). ChIP-Seq analysis was performed by Active Motif, Inc. using an antibody against BRG1 (Abcam) and H3K9Ac (Abcam). Seventy-five-nucleotide reads generated by Illumina sequencing were mapped to the genome using the BWA algorithm with default settings. In total, 26349 peaks were identified over the input control, and the heat maps and average profile for TSS were generated using ngsplot v2.61.

**Immunohistochemistry and transmission electron microscopy.** Tissues were fixed overnight in 4% paraformaldehyde, prepared using the Swiss roll technique, embedded in paraffin, and cut into 7-μm sections. H&E-staining sections were scored by a pathologist in a blinded fashion. Colitis scores were assigned based on a multiplicative index of severity of inflammation (0–3), ulceration (0–3), and hyperplasia of the mucosa (0–4)[59]. To quantify goblet cells, PAS$^+$ cells were quantified from five random fields from at least five mice per genotype. To quantify Paneth cells, lysozyme$^+$ cells were quantified from 100 crypts from at least 3 mice per genotype; data are represented as mean ± SEM. Tumor grades were scored

according to previous report[60]. Briefly, low-grade dysplasia consisted of stratified dysplastic epithelium retaining its columnar shape. The neoplastic crypt cells were elongated and crowded, with hyperchromatic nuclei, but maintained polarity with respect to the basement membrane. High-grade dysplasia was characterized by more pronounced nuclear atypia, with loss of epithelial cell nuclear polarity. The neoplastic glands may show architectural complexities such as a cribriform appearance. For each animal, 10 random lesions (hyperplasia, LG dysplasia or HG dysplasia) were recorded for the quantifications. Thus, the numbers of each sub-type of lesions in each experimental group ($n = 10$ per genotype) were established and the percentages of different subtypes of lesions were compared. The statistical significance between groups was determined by the chi-square test. Primary antibodies used for IHC were as follows: anti-Ki67 (B56), BD Biosciences, Cat# 550609 (1:500); anti-lysosome, DAKO, Cat#A0099 (1:500); anti-8-OHdG, Abcam, Cat# ab48508 (1:500); anti-cleaved caspase-3, Cell Signaling Technology, Cat# 9661 (1:800); anti-E-cadherin, Cell Signaling Technology, Cat#3195T (1:400); and anti-ZO-1, Invitrogen, Cat# 40-2200 (1:500). Biotinylated secondary antibodies were purchased from Jackson Immunology. Staining was visualized with ABC Kit Vectastain Elite (Vector Laboratories) and DAB substrate (Vector Laboratories). PAS staining was performed using PAS staining kits (Sigma-Aldrich) and ALP activity assay was performed using the Alkaline Phosphatase Staining Kit (Vector Laboratories) as described in the manufacturer's protocols. For the analysis of autophagosome by transmission electron microscopy, pieces of colons were fixed at 4 °C in 2.5% glutaraldehyde, post fixed in $OsO_4$ (1%) for 2 h, and then dehydrated through a series of alcohol concentrations, before being progressively embedded in Epon 812 epoxy resin. Micrographs were obtained with a FEI Tecnai G2 Spirit transmission electron microscope.

**RT-qPCR and ChIP-qPCR assays**. RNA was isolated using TRIzol followed by RQ1 RNase-free DNase Set treatment (Promega) according to the manufacturer's instructions. First strand cDNA was synthesized using Superscript II (Invitrogen). SYBR Green Master Mix reagents (Roche) and primer mixtures (Supplementary Table 1) were used for the real-time PCR. Student's t-test was used to statistical analysis and p value <0.05 was considered significant. The ChIP assays were performed using EZ ChIP kit (Millipore). The procedure was as described in the kit provided by the manufacturer. Briefly, isolated IEC cells were fixed by 1% formaldehyde, fragmented by sonication. BRG1 (Abcam, Cat# ab110641) (2 μg 50 μL$^{-1}$) and H3K9Ac (Abcam, Cat# 4441) (2 μg 50 μL$^{-1}$) were then used for immunoprecipitation. After washing and reverse-crosslinking, the precipitated DNA was amplified by primers and quantified by the qPCR. Primer sequences can be found in the Supplementary Table 1.

**Immunoblotting**. The primary antibodies used in this study were as follows: anti-BRG1 (EPNCIR111A) Abcam Cat# ab110641 (1:10,000); anti-Caspase3 (8G10), Cell Signaling Technology, Cat# 9665 (1:1000); anti-C-Caspase 3, Cell Signaling Technology, Cat# 9661 (1:1000); anti-PARP (46D11), Cell Signaling Technology, Cat# 9532 (1:1000); anti-Ki67 (B56), BD Biosciences, Cat# 550609 (1:500); anti-EpCAM (G8.8), BD Biosciences, Cat# 552370 (1:500); anti-ZO1, Invitrogen, Cat# 40-2200 (1:500); anti-8-OHd G, Abcam, Cat# ab48508 (1:500); anti-LC3A/B (D3U4C), Cell Signaling Technology, Cat# 12741 (1:1000); anti-ATG16L1 (D6A5), Cell Signaling Technology, Cat# 8089 (1:1000); anti-PCNA, Santa Cruz Biotechnology, Cat# SC-7909 (1:1000); anti-P-H3, Cell Signaling Technology, Cat# 9701 (1:1000); anti-E-Cadherin, Cell Signaling Technology, Cat# 3195T (1:1000); anti-DCLK1, Abcam, Cat# ab31704 (1:500); anti-Claudin1, Cell Signaling Technology, Cat# 4933T (1:1000); anti-ChgA, Abcam, Cat# ab715 (1:1000); anti-Hes1, Abcam, Cat# ab71559 (1:500). Raw data of immunoblotting can be found in the source data file.

**FACS**. Cell suspensions were subjected to flow cytometry analyses[61]. For gating strategy, first FSC/SSC was applied to gate the live cells and then according to fluorochrome to make the subsequent gates. All the samples in the same experiments and comparisons were gated under the same parameters. The antibodies used in this study were as follows: anti-CD11b-APC, eBioscience, Cat# 47-0112-82 (1:400); anti-F4/80-PE, eBioscience, Cat# 25-4801-82 (1:400); anti-CD4-FITC, eBioscience, Cat# 11-5040-41 (1:400); Gr-1-FITC, eBioscience, Cat# 11-0114-82 (1:400).

**Isolation of lamina propria cells and analysis**. Colons were cut into small pieces and then incubated with RPMI medium supplemented with FBS, 0.5 mM DTT, 5 mM EDTA, and antibiotics at 37 °C for 30 min. After removal of the epithelial layer, the remaining colon segments were incubated at 37 °C with RPMI medium containing 0.5% collagenase D (Roche) and 0.05% DNase (Roche) for 30 min. Lamina propria cells were stained for surface markers CD4, CD11b, F4/80, and Gr-1 using CD4-FITC, CD11b-APC, F4/80-PE, and Gr-1-FITC antibodies, respectively.

**Analysis of ROS signatures and autophagy in patients**. Analysis in human IBD datasets was carried out essentially as previously described[40]. The ROS and BRG1signature were derived from GSE42476 and our own dataset. In order to define the degree of gene signature manifestation within profiles from external

IBD datasets (e.g., GEO GSE11223), we used the previously described t-score metric[40], which was defined for each external profile as the two-sided t-statistic comparing the average of the BRG1-induced genes with the average of the BRG1-repressed genes (genes within the human dataset were first centered to SD from the median of the specimens). For a given dataset, the t-score contrasted the patterns of the BRG1-induced genes with those of the BRG1-repressed genes to derive a single value denoting coordinate expression of the two gene sets. The correlation between BRG1 and ATG16L1, WIPI2b or AMBRA1 mRNA in IBD specimens (GSE57945, $n = 38$) is computed by nonparametric Spearman correlation.

**Statistical analysis**. All experiments were performed using 3–15 mice or at least three independent repeated experiments. Unless otherwise indicated, data presented as the mean ± S.E.M. and statistical significance was determined by a two-tailed Student's t-test. Pearson correlation coefficients were used to evaluate the relationships between BRG1 and gene expressions. $\chi^2$ test were used to determine whether there was a significant difference between the expected frequencies and the observed frequencies in one or more categories. *$p < 0.05$, **$p < 0.01$, and ***$p < 0.001$.

**Reporting summary**. Further information on research design is available in the Nature Research Reporting Summary linked to this article.

## Data availability

The authors declare that all data supporting the findings in this study are available within the paper, Supplementary information and Source data. All data are available from the authors upon reasonable request. RNA-Seq and ChIP-Seq raw data have been deposited in the Gene Expression Omnibus (GEO) under accession number GEO: GSE112128 and GSE135041.

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

## Acknowledgements

This study was supported by funds from Ministry of Science and Technology of the People's Republic of China (2017YFA0102900 to W.Q.G.), National Natural Science Foundation of China (81872406 and 81630073 to W.Q.G., 81772938 to L.L.), State Key Laboratory of Oncogenes and Related Genes (KF01801 to L.L.), Science and Technology Commission of Shanghai Municipality (16JC1405700 to W.Q.G., 18140902700 and 19140905500 to L.L.), High Peak IV fund from Education Commission of Shanghai Municipality on Stem Cell Research (to W.Q.G.), KC Wong foundation (to W.Q.G.). L.L. is supported by Innovation Research Plan from Shanghai Municipal Education Commission (ZXGF082101), and Shanghai Jiao Tong University Medical Engineering Cross Fund (YG2016MS52). The study is also supported by Bio-ID Center, School of Biomedical Engineering, Shanghai Jiao Tong University.

## Author contributions

W.Q.G. and L.L. conceived and designed the experimental approach, and W.Q.G., L.L., and M.L. prepared the manuscript. M.L. and T.S. performed most experiments. N.L. contributed to the computational analysis in the statistical analysis. J.P. and D.F. performed the tissue microarray and pathology analyses. W.L. performed and supervised a specific subset of the experiments and analyses.

## Competing interests

The authors declare no competing interests.
