## [Peer Review File · Nature Communications]

Reviewers' comments:

Reviewer #1 (Remarks to the Author):

This is an excellent study by Liu M et al, exploring the role of BRG1 in colonic inflammation and tumorigenesis. The authors used conditional BRG1 knockout and overexpression mice in combination with Villin-CreERT drivers, and analyzed the colonic (and partly small intestinal) phenotypes with or without DSS and AOM challenge. RNA-seq and ChIP-seq are also performed to pursue mechanistic insights, and the authors speculated that ROS-mediated signaling and autophagic responses may be regulated by BRG1-dependent chromatin remodeling. Although in vivo data are solid and each finding is promising, it does appear that many factors are likely involved in the pathogenesis of colonic inflammation induced by BRG1 deficiency and it remains unclear what is the most critical mechanism that explains the phenotype in BRG1 transgenic animals. Important pieces are still missing in the current manuscript and the authors need to analyze in vivo samples in more detail. Also, in vitro analysis is quite limited and more extensive in vitro organoid assays are highly encouraged. Specific concerns are listed below.

1. Normal expression pattern of BRG1 in intestine and colon. The authors provided some images of BRG1 immunostaining, and it looks like BRG1 is expressed in the lower part of colonic epithelium. But it remains unclear what cell types are normally expressing BRG1 – are these cells mainly proliferating progenitors or basal stem cells (e.g., Lgr5+ cells), and/or some of mature cell types including MUC2+ goblet cells, ChgA+ endocrine cells, Lyz+ Paneth cells, or FABP+ enterocytes? More detailed analysis regarding BRG1 expression (cell types, location etc) is needed.

2. In Fig.1C, the authors quantified the numbers of immune cells in the colon – but how did they quantify? Immunostained images should be provided here. Does CD4+ population include CD4+ Tregs as well? Fig.1D shows upregulation of several cytokines/chemokines in the colon, but where is the source of these gene expression, epithelium or immune cells? Then the authors analyzed PAS staining in the colon and LyZ staining in the intestine in the next panel – but these should be done in both organs. What about MUC2 staining in colon and intestine? Are enteroendocrine cells and tuft cells also reduced in knockouts? These analysis should be performed with or without DSS.

3. Related to point 2, more detailed analysis regarding differentiation status should be analyzed in BRG1 overexpression mice.

4. In Fig.5, the authors somehow try to connect these phenotypes to increased apoptosis and proliferation in knockout animals. However, the time course of these changes is unclear. When do apoptosis and proliferation continuously occur in knockouts following tamoxifen induction, and do they sustain overtime? Also the authors suddenly came up with the idea of ZO-1-mediated tight junction signaling here – but what about other tight junction related protein expression, such as claudins, tubulins, actins, or cadherins?

5. Most likely the authors need to perform intestinal/colon organoid culture experiments in order to look at more direct effects on apoptosis, proliferation, ROS production by BRG1. Method description of IEC culture is minimal and unclear how the authors cultures intestinal epithelial cells. Please use common 3D organoid culture method (Sato T et al) and show/quantify organoid growth, gene changes, apoptosis, ROS, autophagy, etc.

6. Then next the authors focus on autophagic responses and suggest that autophagy is inhibited in knockouts because autophagy-related gene expression is downregulated through BRG1-mediated chromatin remodeling. This is interesting, but what is the sequence of all these changes caused by BRG1 – presumably, loss of BRG1 > impaired autophagy > increased apoptosis and proliferation > impaired barrier function > inflammation > tumorigenesis? Very complicated. The authors only looked at ROS amount (unclear where ROS is involved in the sequential steps above...) and inflammatory cytokines in ATG5 knockout animals, but more detailed analysis is needed to reveal overall pictures.

7. Given that the ROS signaling and autophagy pathway is involved, and that many cytokine-producing immune cells likely play a role here, the authors need to think about the effects by colonic bacterium in all BRG-mediated responses. Does antibiotic treatment improve DSS colitis in knockouts, with recovery of autophagy/ROS/cytokine/apoptosis/barrier function? Any changes in colonic microflora in transgenic animals?

8. It seems odd that there is no change in Lgr5 expression in colon (and intestine too?). Basically, Lgr5+ stem cells are lost during inflammation as reported previously, and as far as inflammation is induced and sustained by BRG1 knockout, one would easily assume that Lgr5 is downregulated in inflamed tissue. The authors double check the Lgr5 expression during the timecourse, perhaps using in situ imaging or Lgr5-GFP animals.

9. Loss of goblet cells and paneth cells suggests Notch activation in knockouts. Although Notch-related RNA expression is not changed, Hes1 IHC analysis should be added here.

Reviewer #2 (Remarks to the Author):

BRG1 is a chromatin remodelling factor that is frequently mutated, or otherwise downregulated, in colitis. This event predisposes to colorectal carcinoma (CRC). Liu et al use genetically modified mice to model BRG1 loss and gain of function, and the impact of these events on development of colitis and early CRC lesions. They show for the first time that high BRG1 levels protect against these pathologies, and vice versa. Mechanistically, they show mechanistically that loss of BRG1 impairs autophagy, as it directly regulates Atg (AuTophagy Gene) transcription in intestinal epithelial cells, leading to ROS accumulation, cell death and loss of intestinal barrier integrity.

While autophagy is already known to be critical for colonic health, this study takes these ideas further by providing an important insight into transcriptional regulation of autophagy in the colon, linking it to a putative colitis (and thus tumour) suppressive gene, BRG1. The study is original and will be of interest to the fields of autophagy and medicine.

The authors acknowledge that other mechanisms for BRG1 influence on colitis may exist, but I agree that, broadly speaking, these are outside the scope of the manuscript, although some descriptive information on whether these are possibilities would be useful in light of data already in the literature (see Minor Points 5 and 6 below). Overall then, the manuscript is well written and the dataset is compelling. I have a small number of outstanding concerns I would ideally see addressed by the authors before I feel I can recommend outright publication:

Major point

1) The study would benefit from some information on how heterozygosity for BRG1 predisposes to basal or DSS-induced colitis and/or DSS+AOM-induced early CRC lesion formation, given that the human data indicates that a downregulation rather than a loss of BRG1 is associated with disease.

Minor points

2) Is BRG1 downregulated upon DSS-induced inflammation in mice?

3) The authors should clarify the background strain of mouse these are using and whether this is inbred or outbred (e.g. C57/BL6J etc.), and the breeding strategy used to produce WT and HOM comparators, e.g. HET x HET matings to produce littermates, or separate WT x WT and HOM x HOM breeding sublines? The strain information is especially important to interpret these studies given the comparison of the results to intestinal adenoma models using Apc mutants (where strain has a big influence).

4) Figure 6d - please provide zoomed and higher resolution images of representative autophagosomes by TEM. I could not definitely identify current resolution images as autophagosomes based upon structural criteria.

5) Given that BRG1 has been reported to directly regulate transcripts involved in oxidative stress responses, such as HO-1, levels of these should be analysed in colons/IECs (descriptive, I don't

think functional probing of this potential mechanism is required).

6) Given that BRG1 has been proposed to regulate mitophagy (specifically mitochondrial clearance by the autophagy), this should be measured in existing samples that were generated from colons and cultured IEC cells.

Simon Wilkinson
University of Edinburgh

Reviewer #3 (Remarks to the Author):

Liu and colleagues undertake a comprehensive study to demonstrate the Brg1 loss enhances ROS production in a cell-intrinsic manner to accelerate barrier dysfunction, inflammation, and colitis in DSS models and enhanced tumorigenesis in AOM/DSS models. This work adds to our expanding knowledge about the role of the SWI/SNF complex in pre-malignant and malignant states, and particularly the context-dependent roles of SWI/SNF subunit deletion. I feel that this work is quite comprehensive as the authors go to great lengths to establish the relationships between Brg1 deletion/overexpression and autophagy regulators using genetic/pharmacologic methods in the DSS model with supporting data in the AOM/DSS model. The data appears to be high quality and substantiated by multiple methods. I appreciate that the authors address previous work, but agree that we have seen context-dependent roles for this complex in several in vivo models, so there are likely many explanations for these discrepancies as the authors point out. I have limited comments to add to this very complete and interesting story.

Main Points:

The BRG1 ChIP is likely underpowered in this study as the authors only pick up 9,500 sites. BRG1 typically binds to several tens of thousands of sites. However, in many cases, BRG1 ChIP-seq data needs to be overlapped with ATAC-seq or histone modification ChIP-seq to uncover sites where BRG1 is required for epigenetic maintenance. The authors perform H3K9ac at their genes of interest. They should perform ATAC-seq or H3K9ac ChIP-seq or H3K27ac ChIP-seq (only one of these would be necessary) in Brg1f/f versus Brg1 IEC-AKO to distinguish between BRG1 binding sites and BRG1 functional sites genome-wide. This would provide an unbiased view of other pathways that BRG1 is regulating in this context. On that note, the authors should list all significant GO terms in Figure 6A.

Do the authors have any insight into how rapamycin may override BRG1 loss? The authors should perform qPCR or RNA-seq or western blot in rapamycin in IEC Brg1f/f and IEC Brg1f/f ERT2 to see if the expression of the autophagy regulators that are typically downregulated in BRG1 deficient cells are rescued.

Minor point:

The authors have misused the word 'ameliorate' in the text where I believe they mean 'enhance', for example:

'We found that Brg1 loss in adult mice leads to ROS accumulations, and thereby ameliorates epithelium death and barrier integrity.'

'Additionally, depletion of Atg5 in the Brg1IEC-OE/+ mice greatly ameliorated the inflammatory cytokine and ROS production to the levels similar as those observed in the Atg5IEC-/- mice (Fig. 7g, h).'

POINT-BY-POINT RESPONSES TO REVIEWERS' COMMENTS

Reviewers' comments:

Reviewer #1 (Remarks to the Author):

This is an excellent study by Liu M et al, exploring the role of BRG1 in colonic inflammation and tumorigenesis. The authors used conditional BRG1 knockout and overexpression mice in combination with Villin-Cre ERT drivers, and analyzed the colonic (and partly small intestinal) phenotypes with or without DSS and AOM challenge. RNA-seq and ChIP-seq are also performed to pursue mechanistic insights, and the authors speculated that ROS-mediated signaling and autophagic responses may be regulated by BRG1-dependent chromatin remodeling. Although in vivo data are solid and each finding is promising, it does appear that many factors are likely involved in the pathogenesis of colonic inflammation induced by BRG1 deficiency and it remains unclear what is the most critical mechanism that explains the phenotype in BRG1 transgenic animals. Important pieces are still missing in the current manuscript and the authors need to analyze in vivo samples in more detail. Also, in vitro analysis is quite limited and more extensive in vitro organoid assays are highly encouraged. Specific concerns are listed below.

Response: We greatly appreciate the reviewer for his/her valuable comments, and we have now revised our manuscript as suggested.

1. Normal expression pattern of BRG1 in intestine and colon. The authors provided some images of BRG1 immunostaining, and it looks like BRG1 is expressed in the lower part of colonic epithelium. But it remains unclear what cell types are normally expressing BRG1 – are these cells mainly proliferating progenitors or basal stem cells (e.g., Lgr5⁺ cells), and/or some of mature cell types including MUC2⁺ goblet cells, ChgA⁺ endocrine cells, Lyz⁺ Paneth cells, or FABP⁺ enterocytes? More detailed analysis regarding BRG1 expression (cell types, location etc) is needed.

Response: We thank the reviewer for pointing out this issue. Actually, BRG1 is uniformly expressed throughout the intestinal epithelium. To verify the expression pattern of BRG1 in intestine and colon, we performed co-immunofluorescence staining of BRG1 with Lgr5, ChgA, Muc2 and Lys respectively in the intestine and colon of wild-type mice. As shown in **supplementary Figure 1A** of the revised manuscript, BRG1⁺ cells were located along the intestinal epithelium and BRG1 was co-expressed with Lgr5, ChgA, Muc2 and Lys, indicating that BRG1 is uniformly expressed both in the basal stem cells and differentiated cells. In addition, to avoid this confusion, we have now updated the images of BRG1 immunostaining in revised **Figure 2A** and **Figure 4B**.

2. In Fig.1C, the authors quantified the numbers of immune cells in the colon – but how did they quantify? Immunostained images should be provided here. Does CD4⁺ population include CD4⁺ Tregs as well? Fig.1D shows upregulation of several cytokines/chemokines in the colon, but where is the source of these gene expression, epithelium or immune cells? Then the authors analyzed PAS staining in the colon and LyZ staining in the intestine in the next panel – but these should be done in both organs. What about MUC2 staining in colon and intestine? Are enteroendocrine cells and tuft cells also reduced in knockouts? These analysis should be performed with or without DSS.

Response: We thank the reviewer for his/her thoughtful comments.

(1) In Figure 2C, we quantified the numbers of CD4⁺ T cells, CD11b⁺; F4/80⁺ macrophages and CD11b⁺; Gr-1⁺ neutrophils of colonic lamina propria by **flow cytometry**. We have now included representative flow cytometry plots in **supplementary Figure 1D** of the revised manuscript. Therefore, CD4⁺ population includes CD4⁺ Treg cells as well.

(2) In Figure 2D, we analyzed the expression levels of cytokines and chemokines in **the whole colon homogenates** that include both epithelial and immune cells.

(3) As suggested by the reviewer, we performed AB-PAS staining in both colons and small intestines, and Lys staining in small intestines. As shown in **Figure 2E** of the revised manuscript, decreased numbers of goblet cells and paneth cells in Brg1^{IEC-AKO} mice were consistently observed. Besides, we also performed Muc2 staining in both colons and small intestines, and found that Muc2 expression was decreased in Brg1^{IEC-AKO} mice (**supplementary Figure 1E**).

(4) As suggested, we also performed ChgA (enteroendocrine cells) and DCLK1 (tuft cells) staining in both colons and small intestines. As shown in **supplementary Figure 1J-K** of the revised manuscript, there was no significant difference in the numbers of ChgA⁺ enteroendocrine cells and DCLK1⁺ tuft cells in Brg1^{IEC-AKO} mice as compared to those in the control mice.

(5) These analyses (same as (3) and (4)) were also performed in Brg1^{F/F} and Brg1^{IEC-AKO} mice with 1% DSS, and the results were similar to (3) and (4) (**supplementary Figure 2** of the revised manuscript).

3. Related to point 2, more detailed analysis regarding differentiation status should be analyzed in BRG1 overexpression mice.

Response: As suggested, we have performed the staining that is the same as response 2 in tissue sections of 3% DSS-treated R26^{Brg/+} and Brg1^{IEC-OE/+} mice and conducted more detailed analysis regarding differentiation status. As shown in **Figure 4G** of the revised manuscript, compared with those in R26^{Brg/+} colons and small intestines, the numbers of goblet cells and Paneth cells were increased in the Brg1^{IEC-OE/+} colons and small intestines. Besides, Muc2 expression was increased in Brg1^{IEC-OE/+} mice compared with that in R26^{Brg/+} mice (**supplementary Figure 3C**). The numbers of ChgA⁺ enteroendocrine cells and DCLK1⁺ tuft cells were not significantly different between R26^{Brg/+} and Brg1^{IEC-OE/+} colons and intestines (**Figure R1** below).

Figure R1. Immunostaining experiments of colons and intestines derived from 3% DSS-treated R26^{Brg1/+} and Brg1^{IEC-OE/+} mice. (A)-(B) Immunostaining of ChgA and DCLK1 in colons and intestines of 3% DSS-treated R26^{Brg1/+} and Brg1^{IEC-OE/+} mice and quantitation results are shown in the right. Scale Bars: 100 μ m **bottom a**, 50 μ m **upper a, b**.

4. In Fig.5, the authors somehow try to connect these phenotypes to increased apoptosis and proliferation in knockout animals. However, the time course of these changes is unclear. When do apoptosis and proliferation continuously occur in knockouts following tamoxifen induction, and do they sustain overtime? Also the authors suddenly came up with the idea of ZO-1-mediated tight junction signaling here — but what about other tight junction related protein expression, such as claudins, tubulins, actins, or cadherins?

Response: We thank the reviewer for his/her thoughtful comments.

(1) In Figure 5, we respectively assessed apoptosis and proliferation in the 10-week-old and 12-week-old Brg1^{IEC-AKO} and Brg1^{F/F} mice. As shown in **supplementary Figure 4B** of the revised manuscript, it did not lead to appreciable changes in terms of cell death and proliferation in 10-week-old (two weeks after tamoxifen treatment) Brg1^{IEC-AKO} mice. However, the 12-week-old (four weeks after tamoxifen treatment) Brg1^{IEC-AKO} mice began to exhibit increased apoptosis and proliferation (**Figure 5C** of the revised manuscript), which were seen in the whole inflammation process.

(2) Considering that intestinal barrier dysfunctions frequently contribute to gut inflammation, we investigated intestinal permeability. Because tight junctions are pivotal in regulating intestinal permeability, we examined the distribution of the tight junction protein 1 (ZO-1) as a marker of tight junction structure in Brg1^{F/F} and Brg1^{IEC-AKO} mice (**Figure 5A** of the revised manuscript). According to the reviewer's suggestion, we have also assessed other junction-related protein expression by immunostaining and western blot. As shown in **Figure 5A and supplementary Figure 4A** of the revised manuscript, the expressions of other junction proteins, such as claudin-1, E-cadherin, α -tubulins and F-actin were also reduced in Brg1^{IEC-AKO} mice. These findings together suggested that Brg1 ablation leads to barrier disruption and colonic leakage.

5. Most likely the authors need to perform intestinal/colon organoid culture experiments in order to look at more direct effects on apoptosis, proliferation, ROS production by BRG1. Method description of IEC culture is minimal and unclear how the authors cultures intestinal epithelial cells. Please use common 3D organoid culture method (Sato T et al) and show/quantify organoid growth, gene changes, apoptosis, ROS, autophagy, etc.

Response: We thank the reviewer for his/her thoughtful comments.

(1) Following the reviewer's advice, we have improved the method description of IEC culture in the revised manuscript (please see our revised Method sections of **IEC isolation and culture**).

(2) As suggested, we have also performed intestinal organoid culture experiments to show organoid growth, apoptosis, autophagy and oxidative stress, which were consistent with the results obtained from mice and IECs experiments in vivo. Briefly, as revealed by the 7-aminoactinomycin D (7-AAD) staining and cleaved caspase-3 immunostaining (**Figure 5E-F** of the revised manuscript), the organoids lacking BRG1 displayed a substantial increase in apoptosis relative to the controls.

Besides, the numbers of Ki67- and 8-OHdG- positive cells were increased in organoids lacking BRG1 (**supplementary Figure 4C** of the revised manuscript). As reflected by L3II conversion and p62 level, Brg1 deletion led to a reduction of autophagy in organoids (**supplementary Figure 5C** of the revised manuscript).

6. Then next the authors focus on autophagic responses and suggest that autophagy is inhibited in knockouts because autophagy-related gene expression is downregulated through BRG1-mediated chromatin remodeling. This is interesting, but what is the sequence of all these changes caused by BRG1- presumably, loss of BRG1 > impaired autophagy > increased apoptosis and proliferation > impaired barrier function > inflammation > tumorigenesis? Very complicated. The authors only looked at ROS amount (unclear where ROS is involved in the sequential steps above...) and inflammatory cytokines in ATG5 knockout animals, but more detailed analysis is needed to reveal overall pictures.

Response: We thank the reviewer for his/her positive comments.

(1) In our study, defective autophagy in BRG1-deficient IECs resulted in excess ROS, which led to the defects in cellular apoptosis and barrier integrity, and the subsequent inflammation and tumorigenesis. Thus, the sequence of all these changes caused by BRG1 loss are presented as below: loss of BRG1 > impaired autophagy > **excess ROS** > increased apoptosis and proliferation > impaired barrier function > inflammation > tumorigenesis.

(2) As suggested, we have performed new experiments and conducted more detailed analyses in ATG5 knockout animals to reveal overall pictures. As shown in **supplementary Figure 6B-E** of the revised manuscript, upon treatment with 3% DSS, compared with the littermate controls, the *Atg5*^{IEC-/-} mice exhibited reduced expressions of ZO-1, claudin1 and E-cadherin and more epithelial apoptosis. More importantly, the *Atg5* ablation in the *Brg1*^{IEC-OE/+} mice led to severe barrier disruption, epithelial cell death and oxidative stress (8-OHdG) comparable to those in the *Atg5*^{IEC-/-} mice. Altogether, these results (**Figure 7** and **supplementary Figure 6**) showed that defective autophagy in *Atg5*-deficient mice resulted in excess ROS, which led to the defects in cellular apoptosis and barrier integrity, and the subsequent inflammation. Thus, Brg1 prevents colon inflammation dependent on the regulation of the autophagy to restrain ROS over-reactions.

7. Given that the ROS signaling and autophagy pathway is involved, and that many cytokine-producing immune cells likely play a role here, the authors need to think about the effects by colonic bacterium in all BRG-mediated responses. Does antibiotic treatment improve DSS colitis in knockouts, with recovery of autophagy/ROS/cytokine/apoptosis/barrier function? Any changes in colonic microflora in transgenic animals?

Response: We thank the reviewer for this valuable question and would like to answer it in two aspects:

(1) To examine whether there are some changes in colonic microflora in *Brg1*^{IEC-AKO} mice, we performed bacterial 16S rDNA sequencing. As shown in **supplementary Figure 7A-C**, there was less bacterial diversity ($p < 0.05$) but unchanged gut microbiota composition in *Brg1*^{IEC-AKO} mice as compared with those in *Brg1*^{F/F} littermates.

(2) To investigate whether antibiotic treatment improves DSS colitis in *Brg1*^{IEC-AKO} mice, we

established gut microbiota-depleted mice by treating $Brg1^{IEC-AKO}$ and $Brg1^{F/F}$ mice with drinking water containing an antibiotic cocktail for 3 weeks and then treated them with 1%DSS. The successful depletion of the gut microbiota was evident as no sufficient bacterial diversity and microbiota composition were observed (**Supplementary Figure 7A-C**). Interestingly, depletion of gut microbiota by antibiotics reduced expression of proinflammatory cytokines and chemokines of $Brg1^{IEC-AKO}$ mice comparable to those observed in the $Brg1^{F/F}$ mice (**Supplementary Figure 7D**). However, antibiotic-treated $Brg1^{IEC-AKO}$ mice still exhibited more severe barrier disruption, epithelial cell apoptosis, impaired autophagy and oxidative stress (8-OHdG) than $Brg1^{F/F}$ mice (**Supplementary Figure 7E-I**). These results suggested that depletion of the gut microbiota did not facilitate the functional recovery of autophagy, apoptosis and intestinal barrier in $Brg1^{IEC-AKO}$ mice. Thus, the gut microbiota does not contribute to barrier integrity caused by BRG1 deficiency. We have added a paragraph on this point in page 15-16 of the revised manuscript text.

8. It seems odd that there is no change in *Lgr5* expression in colon (and intestine too?). Basically, *Lgr5*⁺ stem cells are lost during inflammation as reported previously, and as far as inflammation is induced and sustained by BRG1 knockout, one would easily assume that *Lgr5* is downregulated in inflamed tissue. The authors double check the *Lgr5* expression during the timecourse, perhaps using in situ imaging or *Lgr5*-GFP animals.

Response: According to our data, there is no change in *Lgr5* expression in colon (**Supplementary Figure 1G, H**). To detect *Lgr5* expression in small intestine, we performed additional RT-qPCR analysis which showed no appreciable changes too (**Figure R2A** below). As suggested, we also performed *Lgr5* in situ imaging to double check the *Lgr5* expression during the timecourse. As shown in **Supplementary Figure 1I** (also see **Figure R2B, C** below), *Lgr5* expression levels did not show appreciable changes in both colon and small intestine during the timecourse, which is consistent with that result from RT-qPCR.

Figure R2. Lgr5 expression in colons and intestines of Brg1^{F/F} and Brg1^{IEC-AKO} mice. (A) RT-qPCR analysis of Lgr5 expression in the intestines of Brg1^{F/F} and BRG1^{IEC-AKO} mice at the indicated time points. (B) and (C) Immunofluorescent staining of Lgr5 in colons and intestines of Brg1^{F/F} and BRG1^{IEC-AKO} mice at the indicated time points. Scale Bars: 50 μ m.

9. Loss of goblet cells and paneth cells suggests Notch activation in knockouts. Although Notch-related RNA expression is not changed, Hes1 IHC analysis should be added here.

Response: As suggested, we have performed IHC analysis for Hes1, an Notch signaling effector. As shown in **Supplementary Figure 8C** of the revised manuscript, there still showed no appreciable alternations of Hes1 expression in Brg1-depleted colons.

Reviewer #2 (Remarks to the Author):

BRG1 is a chromatin remodelling factor that is frequently mutated, or otherwise downregulated, in colitis. This event predisposes to colorectal carcinoma (CRC). Liu et al use genetically modified mice to model BRG1 loss and gain of function, and the impact of these events on development of colitis and early CRC lesions. They show for the first time that high BRG1 levels protect against these pathologies, and vice versa. Mechanistically, they show mechanistically that loss of BRG1 impairs autophagy, as it directly regulates Atg (AuTophagy Gene) transcription in intestinal epithelial cells, leading to ROS accumulation, cell death and loss of intestinal barrier integrity.

While autophagy is already known to be critical for colonic health, this study takes these ideas further by providing an important insight into transcriptional regulation of autophagy in the colon, linking it to a putative colitis (and thus tumour) suppressive gene, BRG1. The study is original and will be of interest to the fields of autophagy and medicine.

The authors acknowledge that other mechanisms for BRG1 influence on colitis may exist, but I agree that, broadly speaking, these are outside the scope of the manuscript, although some descriptive information on whether these are possibilities would be useful in light of data already in the literature (see Minor Points 5 and 6 below). Overall then, the manuscript is well written and the dataset is compelling. I have a small number of outstanding concerns I would ideally see addressed by the authors before I feel I can recommend outright publication:

Response: We thank the reviewer very much for the summary and inspiring comments.

1. The study would benefit from some information on how heterozygosity for BRG1 predisposes to basal or DSS-induced colitis and/or DSS+AOM-induced early CRC lesion formation, given that the human data indicates that a downregulation rather than a loss of BRG1 is associated with disease.

Response: To address this concern, we performed DSS-induced colitis experiments on BRG1 heterozygote (Villin^{Cre-ERT2}; Brg1^{flox/+}) mice. As shown in **Figure R3A** below, heterozygote mice showed a similar level of Brg1 protein expression compared with control (Brg1^{flox/flox}) mice. In addition, upon treatment with 3% DSS, heterozygote mice exhibited the same histology and inflammatory response as control mice (**Figure R3B** below). These results suggested that the

inflammation phenotype of heterozygote mice is consistent with control mice.

Figure R3. The phenotype of heterozygote mice for BRG1. (A) Immunoblot (IB) analyses of BRG1 expression are shown. (B) Representative histological images of middle-distal colon tissue collected on day 10 from 3% DSS-treated mice as indicated, and the quantitation of histology score are shown in the right.

2. Is BRG1 downregulated upon DSS-induced inflammation in mice?

Response: Indeed. To answer this question well, we challenged wild-type mice with 3% DSS for 7 days to induce inflammation and evaluated BRG1 expression level 2 days later. As shown in **Figure R4** below, BRG1 is downregulated upon DSS-induced inflammation in mice.

Figure R4. BRG1 expression is decreased in DSS-induced WT mice. (A)-(C) RT-qPCR analysis (A), immunoblotting analysis (B) and immunostaining analysis (C) of BRG1 in the colon tissues as indicated. * $p < 0.05$. Scale Bars: 50 μm c.

3. The authors should clarify the background strain of mouse these are using and whether this is inbred or outbred (e.g. C57/BL6J etc.), and the breeding strategy used to produce WT and HOM comparators, e.g. HET x HET matings to produce littermates, or separate WT x WT and HOM x HOM breeding sublines? The strain information is especially important to interpret these studies given the comparison of the results to intestinal adenoma models using *Apc* mutants (where strain has a big influence).

Response: We thank the reviewer for this kind suggestion. We have now clarified the background information about the mouse strains in the material and methods. All the mice in our study were maintained on C57BL/6J background and littermates with the same treatment were used for control experiments. For example, $Brg1^{IEC-AKO}$ mice (knockout group) and $Brg1^{F/F}$ littermates (control group) were generated by crossing $Villin^{Cre-ERT2}; Brg1^{F/F}$ and $Brg1^{F/F}$ mice.

4. Figure 6d - please provide zoomed and higher resolution images of representative autophagosomes by TEM. I could not definitely identify current resolution images as autophagosomes based upon structural criteria.

Response: As suggested, we have now attached higher resolution images in revised **Figure 6D** (also see **Figure R5** below).

Figure R5. Autophagosome detection by transmission electron microscopy and the quantification in the colons from 7 week-old $Brg1^{F/F}$ and $Brg1^{IEC-AKO}$ mice (One week of $Brg1$ deletion; $n = 5$ per genotype). Arrow indicates the Autophagosome, M: Mitochondrion.

5. Given that BRG1 has been reported to directly regulate transcripts involved in oxidative stress responses, such as HO-1, levels of these should be analysed in colons/IECs (descriptive, I don't think functional probing of this potential mechanism is required).

Response: We thank the review for this helpful suggestion. As requested, we have performed RT-qPCR and Western blot analysis to examine HO-1 expression in colon tissues and IECs from $Brg1^{IEC-AKO}$ and $Brg1^{F/F}$ mice, respectively. Results indicated reduced HO-1 expression in BRG1-deficient mice compared with control groups (**Figure R6 A, B and D** below).

6. Given that BRG1 has been proposed to regulate mitophagy (specifically mitochondrial clearance by the autophagy), this should be measured in existing samples that were generated from colons and cultured IEC cells.

Response: As requested, we have respectively examined BNIP3 and BNIP3L (mitochondrial autophagy markers) expression in colon tissues and IECs from $Brg1^{IEC-AKO}$ and $Brg1^{F/F}$ mice. RT-

qPCR and Western blot analysis indicated reduced BNIP3 and BNIP3L expression in BRG1-deficient mice compared with control groups (Figure R6 A, C and E below).

Figure R6. HO-1, BNIP3 and BNIP3L expression levels. (A) Immunoblotting of HO-1, BNIP3 and BNIP3L levels in Brg1^{IEC-AKO} and Brg1^{F/F} mice. (B) and (C) RT-qPCR analysis of the relative mRNA levels of the indicated genes in whole colonic homogenates from 3-month-old mice. (D) and (E) RT-qPCR analysis of the relative mRNA levels of the indicated genes in IECs from Brg1^{IEC-AKO} and Brg1^{F/F} mice.

Reviewer #3(Remarks to the Author):

Liu and colleagues undertake a comprehensive study to demonstrate the Brg1 loss enhances ROS production in a cell-intrinsic manner to accelerate barrier dysfunction, inflammation, and colitis in DSS models and enhanced tumorigenesis in AOM/DSS models. This work adds to our expanding knowledge about the role of the SWI/SNF complex in pre-malignant and malignant states, and particularly the context-dependent roles of SWI/SNF subunit deletion. I feel that this work is quite comprehensive as the authors go to great lengths to establish the relationships between Brg1 deletion/overexpression and autophagy regulators using genetic/pharmacologic methods in the DSS model with supporting data in the AOM/DSS model. The data appears to be high quality and substantiated by multiple methods. I appreciate that the authors address previous work, but agree that we have seen context-dependent roles for this complex in several in vivo models, so there are likely many explanations for these discrepancies as the authors point out. I have limited comments to add to this very complete and interesting story.

Response: We thank the reviewer very much for the precise summary of our manuscript and positive

comments of our work.

1. The BRG1 ChIP is likely underpowered in this study as the authors only pick up 9,500 sites. BRG1 typically binds to several tens of thousands of sites. However, in many cases, BRG1 ChIP-seq data needs to be overlapped with ATAC-seq or histone modification ChIP-seq to uncover sites where BRG1 is required for epigenetic maintenance. The authors perform H3K9ac at their genes of interest. They should perform ATAC-seq or H3K9ac ChIP-seq or H3K27ac ChIP-seq (only one of these would be necessary) in *Brg1^{f/f}* versus *Brg1^{IEC-AKO}* to distinguish between BRG1 binding sites and BRG1 functional sites genome-wide. This would provide an unbiased view of other pathways that BRG1 is regulating in this context. On that note, the authors should list all significant GO terms in Figure 6A.

Response:

(1) We are sorry that we did not make it clear in the early version of our manuscript. Actually, there were 9504 genes (26349 peaks/sites) identified in the original BRG1 ChIP-seq analysis. We have now re-analyzed the seq data and found that 12119 genes possessed BRG1 occupancies within 6 kb of annotated genes. The Venn diagrams indicated that 746 genes showed direct BRG1 occupancies and an expression downregulation upon *Brg1* ablation (new Figure 6A of the revised manuscript). As requested, we have now listed all significant GO terms in Figure 6A of the revised manuscript (also see Figure R7 A below).

(2) According to the reviewer's advice, we performed H3K9ac ChIP-seq in *Brg1^{F/F}* and *Brg1^{IEC-AKO}* IECs. The ChIP-seq analysis revealed that 12573 genes possessed H3K9ac occupancies within 6 kb of annotated genes (Supplementary Figure 5H, also see Figure R7B below). As the Venn diagrams shown (Figure 6J of the revised manuscript, also see Figure R7 C below), there were still 685 genes overlapped. Among these genes, we noticed that the process named as "Autophagic vacuole fusion", "Autophagic vacuole assembly" and "Autophagy" was still significantly enriched (Supplementary Figure 5I of the revised manuscript, also see Figure R7 C below). In addition, we also included the snapshot of H3K9ac ChIP-Seq signals at the *Atg16l1*, *Wipi2b*, *Atg7* and *Ambra1* gene loci in IECs isolated from *Brg1^{F/F}* and *Brg1^{IEC-AKO}* mice in revised Supplementary Figure 5 J (also see Figure R7 D below). H3K9ac recruitment to these autophagy-regulating gene loci was simultaneously reduced in the *Brg1*-depleted IECs. These results strongly suggest that BRG1 modulates the multifaceted autophagy machinery to govern colon homeostasis.

Figure R7. ChIP-Seq and RNA-Seq analyses. (A) Venn diagram showing the number of genes harboring BRG1 binding and displaying expression changes in Brg1-KO IECs (One week of Brg1 ablation). Right panel shows the significant enrichments of GO terms in the overlapping genes. (B) Average H3K9ac ChIP signal across 12573 annotated genes in IECs isolated from Brg1^{F/F} and Brg1^{IEC-AKO} mice. (C) Venn diagram indicating overlapping genes with BRG1, H3K9ac binding and displaying expression changes in Brg1-KO IECs. Right panel shows the significant enrichments of GO terms in the overlapping genes. (D) Snapshot of H3K9ac ChIP-Seq signals at the *Atg16l1*, *Wipi2*, *Atg7* and *Ambra1* gene loci in IECs isolated from Brg1^{F/F} and Brg1^{IEC-AKO} mice.

2. Do the authors have any insight into how rapamycin may override BRG1 loss? The authors should

perform qPCR or RNA-seq or western blot in rapamycin in IEC Brg1^{f/f} and IEC Brg1^{f/f} ERT2 to see if the expression of the autophagy regulators that are typically downregulated in BRG1 deficient cells are rescued.

Response: As suggested, we have performed rapamycin rescue experiment and RT-qPCR on key autophagy regulators in IEC^{Brg1-F/F} and IEC^{Brg1-F/F; CreERT2}. As shown in **Figure R8**, treatment with rapamycin enhanced the expression levels of the autophagy regulators in Brg1-depleted IECs comparable to those observed in the IEC^{Brg1-F/F} cells. The new data has been included to **Supplementary Figure 5G** of the revise manuscript.

Figure R8. RT-qPCR analysis of the relative mRNA levels of the indicated genes in IEC^{Brg1-F/F} and IEC^{Brg1-F/F; CreERT2} with or without 16 hours of rapamycin treatment. (n=5 per genotype). * p < 0.05; ** p < 0.01.

3. The authors have misused the word 'ameliorate' in the text where I believe they mean 'enhance', for example:

'We found that Brg1 loss in adult mice leads to ROS accumulations, and thereby ameliorates epithelium death and barrier integrity.'

'Additionally, depletion of Atg5 in the Brg1IEC-OE/+ mice greatly ameliorated the inflammatory cytokine and ROS production to the levels similar as those observed in the Atg5IEC-/- mice (Fig. 7g, h).'

Response: We are very grateful to the reviewer's careful reading and pointing out the problem. We have corrected the word 'ameliorate' and replaced with 'enhance' in the revised manuscript.

REVIEWERS' COMMENTS:

Reviewer #1 (Remarks to the Author):

Most points raised in the previous review were addressed and now the manuscript is improved. However, the answer regarding to Lgr5 expression (point 8) is not convincing. First, the authors need to compare the Lgr5 expression level between the time points. The authors now set the control mice at each time point as 1.0 and show relative expression, but is the Lgr5 expression at 4 week equivalent to that at 2 week or untreated mice? In general, Lgr5 expression must decrease during inflammation. And the authors used Lg5 immunofluorescence. But there is no quantitation here - please quantify the number of Lgr5+ cells per gland. Also, Lgr5 staining is usually very hard and unreliable. Please consider to perform in situ imaging.

Reviewer #2 (Remarks to the Author):

The authors have satisfied all my queries/concerns pertaining to the original manuscript adequately. I have no further comments that require address. The manuscript remains interesting and timely.

Reviewer #3 (Remarks to the Author):

The authors have addressed my concerns. I would like to commend the authors on a very complete story.

POINT-BY-POINT RESPONSES TO REVIEWERS' COMMENTS

REVIEWERS' COMMENTS:

Reviewer #1 (Remarks to the Author):

Most points raised in the previous review were addressed and now the manuscript is improved.

However, the answer regarding to Lgr5 expression (point 8) is not convincing. First, the authors need to compare the Lgr5 expression level between the time points. The authors now set the control mice at each time point as 1.0 and show relative expression, but is the Lgr5 expression at 4 week equivalent to that at 2 week or untreated mice? In general, Lgr5 expression must decrease during inflammation. And the authors used Lg5 immunofluorescence. But there is no quantitation here - please quantify the number of Lgr5+ cells per gland. Also, Lgr5 staining is usually very hard and unreliable. Please consider to perform *in situ* imaging.

Response: We are very happy to hear the positive comments from the reviewer. To better address the concern of Lgr5 expression, we expanded sample size and repeated RT-qPCR and *in situ* staining experiment of Lgr5. The Lgr5 expression levels were re-analyzed between the time points.

- (1) As suggested, we performed **new RT-qPCR analysis** of Lgr5 during the timecourse (expanded the number of mice to n=8 per genotype) and compared the Lgr5 mRNA expression level between the time points. As shown in **Figure R1A** below, there is still no appreciable change in Lgr5 mRNA expression in both colons and small intestines between the time points.
- (2) As suggested, we also performed **new Lgr5 *in situ* staining experiment** to double check the Lgr5 protein expression during the timecourse (expanded the number of mice to n=12 per genotype) and provided quantitation results in the right of staining images. As shown in **Supplementary Figure 1I** (also see **Figure R1B** below), 2 weeks of Brg1 ablation did not lead to appreciable changes in terms of Lgr5+ cells in both colons and small intestines. However, the 12-week-old Brg1^{IEC-AKO} mice (4 weeks of Brg1 ablation) began to exhibit **decreases in the numbers of Lgr5+ cells** in both colons and small intestines.

Figure R1. Lgr5 expression in colons and intestines of Brg1^{F/F} and Brg1^{IEC-AKO} mice. (A) RT-qPCR analysis of Lgr5 expression in colons and intestines of Brg1^{F/F} and Brg1^{IEC-AKO} mice at the indicated time points (n=8 per genotype). (B) Immunofluorescent staining of Lgr5 in colons and intestines of Brg1^{F/F} and Brg1^{IEC-AKO} mice at the indicated time points (n=12 per genotype). Scale Bars: 50 μ m. * p < 0.05. N.S. : Not Significant, TM: Tamoxifen, W: Week.

Reviewer #2 (Remarks to the Author):

The authors have satisfied all my queries/concerns pertaining to the original manuscript adequately. I have no further comments that require address. The manuscript remains interesting and timely.

Response: We thank the reviewer for the encouraging and positive comments.

Reviewer #3 (Remarks to the Author):

The authors have addressed my concerns. I would like to commend the authors on a very complete story.

Response: We thank the reviewer for the positive comments.